# Hindsight policy gradients

**Paulo Rauber**
IDSIA, USI, SUPSI
Lugano, Switzerland
`paulo@idsia.ch`

**Avinash Ummadisingu**
USI
Lugano, Switzerland
`avinash.ummadisingu@usi.ch`

**Filipe Mutz** [*]
IFES, UFES
Serra, Brazil
`filipe.mutz@ifes.edu.br`

**Jürgen Schmidhuber**
IDSIA, USI, SUPSI, NNAISENSE
Lugano, Switzerland
`juergen@idsia.ch`

## Abstract

A reinforcement learning agent that needs to pursue different goals across episodes requires a goal-conditional policy. In addition to their potential to generalize desirable behavior to unseen goals, such policies may also enable higher-level planning based on subgoals. In sparse-reward environments, the capacity to exploit information about the degree to which an arbitrary goal has been achieved while another goal was intended appears crucial to enable sample efficient learning. However, reinforcement learning agents have only recently been endowed with such capacity for hindsight. In this paper, we demonstrate how hindsight can be introduced to policy gradient methods, generalizing this idea to a broad class of successful algorithms. Our experiments on a diverse selection of sparse-reward environments show that hindsight leads to a remarkable increase in sample efficiency.

## 1 Introduction

In a traditional reinforcement learning setting, an agent interacts with an environment in a sequence of episodes, observing states and acting according to a policy that ideally maximizes expected cumulative reward. If an agent is required to pursue different *goals* across episodes, its *goal-conditional policy* may be represented by a probability distribution over actions for every combination of state and goal. This distinction between states and goals is particularly useful when the probability of a state transition given an action is independent of the goal pursued by the agent.

Learning such goal-conditional behavior has received significant attention in machine learning and robotics, especially because a goal-conditional policy may generalize desirable behavior to goals that were never encountered by the agent (Schmidhuber & Huber, 1990; Da Silva et al., 2012; Kupcsik et al., 2013; Deisenroth et al., 2014; Schaul et al., 2015; Zhu et al., 2017; Kober et al., 2012; Ghosh et al., 2018; Mankowitz et al., 2018; Pathak et al., 2018). Consequently, developing goal-based curricula to facilitate learning has also attracted considerable interest (Fabisch & Metzen, 2014; Florensa et al., 2017; Sukhbaatar et al., 2018; Srivastava et al., 2013). In hierarchical reinforcement learning, goal-conditional policies may enable agents to plan using subgoals, which abstracts the details involved in lower-level decisions (Oh et al., 2017; Vezhnevets et al., 2017; Kulkarni et al., 2016; Levy et al., 2017).

In a typical *sparse-reward environment*, an agent receives a non-zero reward only upon reaching a *goal state*. Besides being natural, this task formulation avoids the potentially difficult problem of *reward shaping*, which often biases the learning process towards suboptimal behavior (Ng et al., 1999). Unfortunately, sparse-reward environments remain particularly challenging for traditional reinforcement learning algorithms (Andrychowicz et al., 2017; Florensa et al., 2017). For example, consider an agent tasked with traveling between cities. In a sparse-reward formulation, if reaching a desired destination by chance is unlikely, a learning agent will rarely obtain reward signals. At

---

[*]Work performed while at IDSIA.

the same time, it seems natural to expect that an agent will learn how to reach the cities it visited regardless of its desired destinations.

In this context, the capacity to exploit information about the degree to which an arbitrary goal has been achieved while another goal was intended is called *hindsight*. This capacity was recently introduced by Andrychowicz et al. (2017) to off-policy reinforcement learning algorithms that rely on experience replay (Lin, 1992). In earlier work, Karkus et al. (2016) introduced hindsight to policy search based on Bayesian optimization (Metzen et al., 2015).

In this paper, we demonstrate how hindsight can be introduced to policy gradient methods (Williams, 1986; 1992; Sutton et al., 1999a), generalizing this idea to a successful class of reinforcement learning algorithms (Peters & Schaal, 2008; Duan et al., 2016).

In contrast to previous work on hindsight, our approach relies on *importance sampling* (Bishop, 2013). In reinforcement learning, importance sampling has been traditionally employed in order to efficiently reuse information obtained by earlier policies during learning (Precup et al., 2000; Peshkin & Shelton, 2002; Jie & Abbeel, 2010; Thomas et al., 2015; Munos et al., 2016). In comparison, our approach attempts to efficiently learn about different goals using information obtained by the current policy for a specific goal. This approach leads to multiple formulations of a *hindsight policy gradient* that relate to well-known policy gradient results.

In comparison to conventional (goal-conditional) policy gradient estimators, our proposed estimators lead to remarkable sample efficiency on a diverse selection of sparse-reward environments.

## 2 PRELIMINARIES

We denote random variables by upper case letters and assignments to these variables by corresponding lower case letters. We let $\text{Val}(X)$ denote the set of valid assignments to a random variable $X$. We also omit the subscript that typically relates a probability function to random variables when there is no risk of ambiguity. For instance, we may use $p(x)$ to denote $p_X(x)$ and $p(y)$ to denote $p_Y(y)$.

Consider an agent that interacts with its environment in a sequence of episodes, each of which lasts for exactly $T$ time steps. The agent receives a goal $g \in \text{Val}(G)$ at the beginning of each episode. At every time step $t$, the agent observes a state $s_t \in \text{Val}(S_t)$, receives a reward $r(s_t, g) \in \mathbb{R}$, and chooses an action $a_t \in \text{Val}(A_t)$. For simplicity of notation, suppose that $\text{Val}(G), \text{Val}(S_t)$, and $\text{Val}(A_t)$ are finite for every $t$.

In our setting, a goal-conditional policy defines a probability distribution over actions for every combination of state and goal. The same policy is used to make decisions at every time step.

Let $\boldsymbol{\tau} = s_1, a_1, s_2, a_2, \ldots, s_{T-1}, a_{T-1}, s_T$ denote a trajectory. We assume that the probability $p(\boldsymbol{\tau} \mid g, \boldsymbol{\theta})$ of trajectory $\boldsymbol{\tau}$ given goal $g$ and a policy parameterized by $\boldsymbol{\theta} \in \text{Val}(\boldsymbol{\Theta})$ is given by

$$p(\boldsymbol{\tau} \mid g, \boldsymbol{\theta}) = p(s_1) \prod_{t=1}^{T-1} p(a_t \mid s_t, g, \boldsymbol{\theta}) p(s_{t+1} \mid s_t, a_t). \tag{1}$$

In contrast to a Markov decision process, this formulation allows the probability of a state transition given an action to change across time steps within an episode. More importantly, it implicitly states that the probability of a state transition given an action is independent of the goal pursued by the agent, which we denote by $S_{t+1} \perp\!\!\!\perp G \mid S_t, A_t$. For every $\boldsymbol{\tau}, g$, and $\boldsymbol{\theta}$, we also assume that $p(\boldsymbol{\tau} \mid g, \boldsymbol{\theta})$ is non-zero and differentiable with respect to $\boldsymbol{\theta}$.

Assuming that $G \perp\!\!\!\perp \boldsymbol{\Theta}$, the expected return $\eta(\boldsymbol{\theta})$ of a policy parameterized by $\boldsymbol{\theta}$ is given by

$$\eta(\boldsymbol{\theta}) = \mathbb{E}\left[\sum_{t=1}^{T} r(S_t, G) \mid \boldsymbol{\theta}\right] = \sum_{g} p(g) \sum_{\boldsymbol{\tau}} p(\boldsymbol{\tau} \mid g, \boldsymbol{\theta}) \sum_{t=1}^{T} r(s_t, g). \tag{2}$$

The action-value function is given by $Q_t^{\boldsymbol{\theta}}(s, a, g) = \mathbb{E}\left[\sum_{t'=t+1}^{T} r(S_{t'}, g) \mid S_t = s, A_t = a, g, \boldsymbol{\theta}\right]$, the value function by $V_t^{\boldsymbol{\theta}}(s, g) = \mathbb{E}\left[Q_t^{\boldsymbol{\theta}}(s, A_t, g) \mid S_t = s, g, \boldsymbol{\theta}\right]$, and the advantage function by $A_t^{\boldsymbol{\theta}}(s, a, g) = Q_t^{\boldsymbol{\theta}}(s, a, g) - V_t^{\boldsymbol{\theta}}(s, g)$.

## 3 GOAL-CONDITIONAL POLICY GRADIENTS

This section presents results for goal-conditional policies that are analogous to well-known results for conventional policies (Peters & Schaal, 2008). They establish the foundation for the results presented in the next section. The corresponding proofs are included in Appendix A for completeness.

The objective of policy gradient methods is finding policy parameters that achieve maximum expected return. When combined with Monte Carlo techniques (Bishop, 2013), the following result allows pursuing this objective using gradient-based optimization.

**Theorem 3.1** (Goal-conditional policy gradient). *The gradient $\nabla \eta(\boldsymbol{\theta})$ of the expected return with respect to $\boldsymbol{\theta}$ is given by*

$$\nabla \eta(\boldsymbol{\theta}) = \sum_g p(g) \sum_{\boldsymbol{\tau}} p(\boldsymbol{\tau} \mid g, \boldsymbol{\theta}) \sum_{t=1}^{T-1} \nabla \log p(a_t \mid s_t, g, \boldsymbol{\theta}) \sum_{t'=t+1}^{T} r(s_{t'}, g). \quad (3)$$

The following result allows employing a *baseline* to reduce the variance of the gradient estimator.

**Theorem 3.2** (Goal-conditional policy gradient, baseline formulation). *For every $t, \boldsymbol{\theta}$, and associated real-valued (baseline) function $b_t^{\boldsymbol{\theta}}$, the gradient $\nabla \eta(\boldsymbol{\theta})$ of the expected return with respect to $\boldsymbol{\theta}$ is given by*

$$\nabla \eta(\boldsymbol{\theta}) = \sum_g p(g) \sum_{\boldsymbol{\tau}} p(\boldsymbol{\tau} \mid g, \boldsymbol{\theta}) \sum_{t=1}^{T-1} \nabla \log p(a_t \mid s_t, g, \boldsymbol{\theta}) \left[ \left[ \sum_{t'=t+1}^{T} r(s_{t'}, g) \right] - b_t^{\boldsymbol{\theta}}(s_t, g) \right]. \quad (4)$$

Appendix A.7 presents the constant baselines that minimize the (elementwise) variance of the corresponding estimator. However, such baselines are usually impractical to compute (or estimate), and the variance of the estimator may be reduced further by a baseline function that depends on state and goal. Although generally suboptimal, it is typical to let the baseline function $b_t^{\boldsymbol{\theta}}$ approximate the value function $V_t^{\boldsymbol{\theta}}$ (Greensmith et al., 2004).

Lastly, actor-critic methods may rely on the following result for goal-conditional policies.

**Theorem 3.3** (Goal-conditional policy gradient, advantage formulation). *The gradient $\nabla \eta(\boldsymbol{\theta})$ of the expected return with respect to $\boldsymbol{\theta}$ is given by*

$$\nabla \eta(\boldsymbol{\theta}) = \sum_g p(g) \sum_{\boldsymbol{\tau}} p(\boldsymbol{\tau} \mid g, \boldsymbol{\theta}) \sum_{t=1}^{T-1} \nabla \log p(a_t \mid s_t, g, \boldsymbol{\theta}) A_t^{\boldsymbol{\theta}}(s_t, a_t, g). \quad (5)$$

## 4 HINDSIGHT POLICY GRADIENTS

This section presents the novel ideas that introduce hindsight to policy gradient methods. The corresponding proofs can be found in Appendix B.

Suppose that the reward $r(s, g)$ is known for every combination of state $s$ and goal $g$, as in previous work on hindsight (Andrychowicz et al., 2017; Karkus et al., 2016). In that case, it is possible to evaluate a trajectory obtained while trying to achieve an original goal $g'$ for an alternative goal $g$. Using importance sampling, this information can be exploited using the following central result.

**Theorem 4.1** (Every-decision hindsight policy gradient). *For an arbitrary (original) goal $g'$, the gradient $\nabla \eta(\boldsymbol{\theta})$ of the expected return with respect to $\boldsymbol{\theta}$ is given by*

$$\nabla \eta(\boldsymbol{\theta}) = \sum_{\boldsymbol{\tau}} p(\boldsymbol{\tau} \mid g', \boldsymbol{\theta}) \sum_g p(g) \sum_{t=1}^{T-1} \nabla \log p(a_t \mid s_t, g, \boldsymbol{\theta}) \sum_{t'=t+1}^{T} \left[ \prod_{k=1}^{T-1} \frac{p(a_k \mid s_k, g, \boldsymbol{\theta})}{p(a_k \mid s_k, g', \boldsymbol{\theta})} \right] r(s_{t'}, g). \quad (6)$$

In the formulation presented above, every reward is multiplied by the ratio between the likelihood of the corresponding trajectory under an alternative goal and the likelihood under the original goal (see Eq. 1). Intuitively, every reward should instead be multiplied by a *likelihood ratio* that only considers the corresponding trajectory up to the previous action. This intuition underlies the following important result, named after an analogous result for action-value functions by Precup et al. (2000).

**Theorem 4.2** (Per-decision hindsight policy gradient). *For an arbitrary (original) goal $g'$, the gradient $\nabla \eta(\boldsymbol{\theta})$ of the expected return with respect to $\boldsymbol{\theta}$ is given by*

$$\nabla \eta(\boldsymbol{\theta}) = \sum_{\boldsymbol{\tau}} p(\boldsymbol{\tau} \mid g', \boldsymbol{\theta}) \sum_{g} p(g) \sum_{t=1}^{T-1} \nabla \log p(a_t \mid s_t, g, \boldsymbol{\theta}) \sum_{t'=t+1}^{T} \left[ \prod_{k=1}^{t'-1} \frac{p(a_k \mid s_k, g, \boldsymbol{\theta})}{p(a_k \mid s_k, g', \boldsymbol{\theta})} \right] r(s_{t'}, g). \quad (7)$$

The following lemma allows introducing baselines to hindsight policy gradients (see App. B.4).

**Lemma 4.1.** *For every $g'$, $t$, $\boldsymbol{\theta}$, and associated real-valued (baseline) function $b_t^{\boldsymbol{\theta}}$,*

$$\sum_{\boldsymbol{\tau}} p(\boldsymbol{\tau} \mid g', \boldsymbol{\theta}) \sum_{g} p(g) \sum_{t=1}^{T-1} \nabla \log p(a_t \mid s_t, g, \boldsymbol{\theta}) \left[ \prod_{k=1}^{t} \frac{p(a_k \mid s_k, g, \boldsymbol{\theta})}{p(a_k \mid s_k, g', \boldsymbol{\theta})} \right] b_t^{\boldsymbol{\theta}}(s_t, g) = \mathbf{0}. \quad (8)$$

Appendix B.7 presents the constant baselines that minimize the (elementwise) variance of the corresponding gradient estimator. By analogy with the conventional practice, we suggest letting the baseline function $b_t^{\boldsymbol{\theta}}$ approximate the value function $V_t^{\boldsymbol{\theta}}$ instead.

Importantly, the choice of likelihood ratio in Lemma 4.1 is far from unique. However, besides leading to straightforward estimation, it also underlies the advantage formulation presented below.

**Theorem 4.3** (Hindsight policy gradient, advantage formulation). *For an arbitrary (original) goal $g'$, the gradient $\nabla \eta(\boldsymbol{\theta})$ of the expected return with respect to $\boldsymbol{\theta}$ is given by*

$$\nabla \eta(\boldsymbol{\theta}) = \sum_{\boldsymbol{\tau}} p(\boldsymbol{\tau} \mid g', \boldsymbol{\theta}) \sum_{g} p(g) \sum_{t=1}^{T-1} \nabla \log p(a_t \mid s_t, g, \boldsymbol{\theta}) \left[ \prod_{k=1}^{t} \frac{p(a_k \mid s_k, g, \boldsymbol{\theta})}{p(a_k \mid s_k, g', \boldsymbol{\theta})} \right] A_t^{\boldsymbol{\theta}}(s_t, a_t, g). \quad (9)$$

Fortunately, the following result allows approximating the advantage under a goal using a state transition collected while pursuing another goal (see App. D.4).

**Theorem 4.4.** *For every $t$ and $\boldsymbol{\theta}$, the advantage function $A_t^{\boldsymbol{\theta}}$ is given by*

$$A_t^{\boldsymbol{\theta}}(s, a, g) = \mathbb{E}\left[ r(S_{t+1}, g) + V_{t+1}^{\boldsymbol{\theta}}(S_{t+1}, g) - V_t^{\boldsymbol{\theta}}(s, g) \mid S_t = s, A_t = a \right]. \quad (10)$$

## 5 HINDSIGHT GRADIENT ESTIMATORS

This section details gradient estimation based on the results presented in the previous section. The corresponding proofs can be found in Appendix C.

Consider a dataset (batch) $\mathcal{D} = \{(\boldsymbol{\tau}^{(i)}, g^{(i)})\}_{i=1}^{N}$ where each trajectory $\boldsymbol{\tau}^{(i)}$ is obtained using a policy parameterized by $\boldsymbol{\theta}$ in an attempt to achieve a goal $g^{(i)}$ chosen by the environment.

The following result points to a straightforward estimator based on Theorem 4.2.

**Theorem 5.1.** *The per-decision hindsight policy gradient estimator, given by*

$$\frac{1}{N} \sum_{i=1}^{N} \sum_{g} p(g) \sum_{t=1}^{T-1} \nabla \log p(A_t^{(i)} \mid S_t^{(i)}, G^{(i)} = g, \boldsymbol{\theta}) \sum_{t'=t+1}^{T} \left[ \prod_{k=1}^{t'-1} \frac{p(A_k^{(i)} \mid S_k^{(i)}, G^{(i)} = g, \boldsymbol{\theta})}{p(A_k^{(i)} \mid S_k^{(i)}, G^{(i)}, \boldsymbol{\theta})} \right] r(S_{t'}^{(i)}, g),$$

$$\quad (11)$$

*is a consistent and unbiased estimator of the gradient $\nabla \eta(\boldsymbol{\theta})$ of the expected return.*

In preliminary experiments, we found that this estimator leads to unstable learning progress, which is probably due to its potential high variance. The following result, inspired by weighted importance sampling (Bishop, 2013), represents our attempt to trade variance for bias.

**Theorem 5.2.** *The weighted per-decision hindsight policy gradient estimator, given by*

$$\sum_{i=1}^{N} \sum_{g} p(g) \sum_{t=1}^{T-1} \nabla \log p(A_t^{(i)} \mid S_t^{(i)}, G^{(i)} = g, \boldsymbol{\theta}) \sum_{t'=t+1}^{T} \frac{\left[ \prod_{k=1}^{t'-1} \frac{p(A_k^{(i)} \mid S_k^{(i)}, G^{(i)} = g, \boldsymbol{\theta})}{p(A_k^{(i)} \mid S_k^{(i)}, G^{(i)}, \boldsymbol{\theta})} \right] r(S_{t'}^{(i)}, g)}{\sum_{j=1}^{N} \left[ \prod_{k=1}^{t'-1} \frac{p(A_k^{(j)} \mid S_k^{(j)}, G^{(j)} = g, \boldsymbol{\theta})}{p(A_k^{(j)} \mid S_k^{(j)}, G^{(j)}, \boldsymbol{\theta})} \right]}, \quad (12)$$

*is a consistent estimator of the gradient $\nabla \eta(\boldsymbol{\theta})$ of the expected return.*

In simple terms, the likelihood ratio for every combination of trajectory, (alternative) goal, and time step is normalized across trajectories by this estimator. In Appendix C.3, we present a result that enables the corresponding consistency-preserving *weighted baseline*.

Consider a set $\mathcal{G}^{(i)} = \{g \in \text{Val}(G) \mid \text{exists a } t \text{ such that } r(s_t^{(i)}, g) \neq 0\}$ composed of so-called *active goals* during the $i$-th episode. The feasibility of the proposed estimators relies on the fact that only active goals correspond to non-zero terms inside the expectation over goals in Expressions 11 and 12. In many natural sparse-reward environments, active goals will correspond directly to states visited during episodes (for instance, the cities visited while trying to reach other cities), which enables computing said expectation exactly when the goal distribution is known.

The proposed estimators have remarkable properties that differentiate them from previous (weighted) importance sampling estimators for off-policy learning. For instance, although a trajectory is often more likely under the original goal than under an alternative goal, in policies with strong optimal substructure, a high probability of a trajectory between the state $a$ and the goal (state) $c$ that goes through the state $b$ may naturally allow for a high probability of the corresponding (sub)trajectory between the state $a$ and the goal (state) $b$. In other cases, the (unnormalized) likelihood ratios may become very small for some (alternative) goals after a few time steps across all trajectories. After normalization, in the worst case, this may even lead to equivalent ratios for such goals for a given time step across all trajectories. In any case, it is important to note that only likelihood ratios associated to active goals for a given episode will affect the gradient estimate. Additionally, an original goal will always have (unnormalized) likelihood ratios equal to one for the corresponding episode.

Under mild additional assumptions, the proposed estimators also allow using a dataset containing goals chosen arbitrarily (instead of goals drawn from the goal distribution). Although this feature is not required by our experiments, we believe that it may be useful to circumvent *catastrophic forgetting* during curriculum learning (McCloskey & Cohen, 1989; Kirkpatrick et al., 2017).

# 6 EXPERIMENTS

This section reports results of an empirical comparison between goal-conditional policy gradient estimators and hindsight policy gradient estimators.[1] Because there are no well-established sparse-reward environments intended to test agents under multiple goals, this comparison focuses on our own selection of environments. These environments are diverse in terms of stochasticity, state space dimensionality and size, relationship between goals and states, and number of actions. In every one of these environments, the agent receives the remaining number of time steps plus one as a reward for reaching the goal state, which also ends the episode. In every other situation, the agent receives no reward.

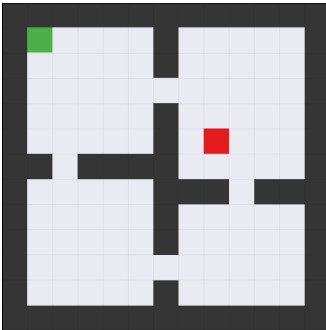
Figure 1: Four rooms.

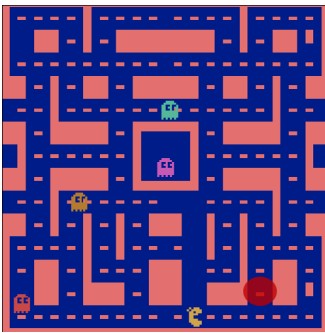
Figure 2: Ms. Pac-man.

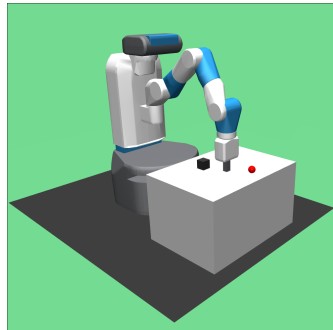
Figure 3: FetchPush.

Importantly, the weighted per-decision hindsight policy gradient estimator used in our experiments (*HPG*) does not precisely correspond to Expression 12. Firstly, the original estimator requires a constant number of time steps $T$, which would often require the agent to act *after* the end of an episode in the environments that we consider. Secondly, although it is feasible to compute Expression

---

[1]An open-source implementation of these estimators is available on http://paulorauber.com/hpg.

12 exactly when the goal distribution is known (as explained in Sec. 5), we sometimes subsample the sets of active goals per episode. Furthermore, when including a baseline that approximates the value function, we again consider only active goals, which by itself generally results in an inconsistent estimator (*HPG+B*). As will become evident in the following sections, these *compromised* estimators still lead to remarkable sample efficiency.

We assess sample efficiency through *learning curves* and *average performance* scores, which are obtained as follows. After collecting a number of batches (composed of trajectories and goals), each of which enables one step of gradient ascent, an agent undergoes *evaluation*. During evaluation, the agent interacts with the environment for a number of episodes, selecting actions with maximum probability according to its policy. A learning curve shows the average return obtained during each evaluation step, averaged across multiple *runs* (independent learning procedures). The curves presented in this text also include a $95\%$ bootstrapped confidence interval. The average performance is given by the average return across evaluation steps, averaged across runs. During both training and evaluation, goals are drawn uniformly at random. Note that there is no held-out set of goals for evaluation, since we are interested in evaluating sample efficiency instead of generalization.

For every combination of environment and batch size, grid search is used to select hyperparameters for each estimator according to average performance scores (after the corresponding standard deviation across runs is subtracted, as suggested by Duan et al. (2016)). *Definitive results* are obtained by using the best hyperparameters found for each estimator in additional runs. In this section, we discuss definitive results for small (2) and medium (16) batch sizes.

More details about our experiments can be found in Appendices E.1 and E.2. Appendix E.3 contains unabridged results, a supplementary empirical study of likelihood ratios (Appendix E.3.6), and an empirical comparison with hindsight experience replay (Appendix E.3.7).

## 6.1 BIT FLIPPING ENVIRONMENTS

In a *bit flipping* environment, the agent starts every episode in the same state (**0**, represented by $k$ bits), and its goal is to reach a randomly chosen state. The actions allow the agent to toggle (flip) each bit individually. The maximum number of time steps is $k + 1$. Despite its apparent simplicity, this environment is an ideal testbed for reinforcement learning algorithms intended to deal with sparse rewards, since obtaining a reward by chance is unlikely even for a relatively small $k$. Andrychowicz et al. (2017) employed a similar environment to evaluate their hindsight approach.

Figure 4 presents the learning curves for $k = 8$. Goal-conditional policy gradient estimators with and without an approximate value function baseline (*GCPG+B* and *GCPG*, respectively) obtain excellent policies and lead to comparable sample efficiency. HPG+B obtains excellent policies more than $400$ batches earlier than these estimators, but its policies degrade upon additional training. Additional experiments strongly suggest that the main cause of this issue is the fact that the value function baseline is still very poorly fit by the time that the policy exhibits desirable behavior. In comparison, *HPG* obtains excellent policies as early as HPG+B, but its policies remain remarkably stable upon additional training.

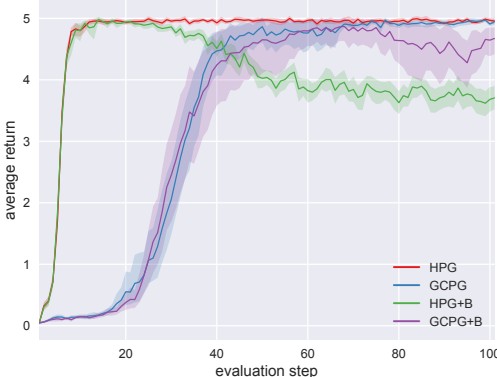

Figure 4: Bit flipping ($k = 8$, batch size 16).

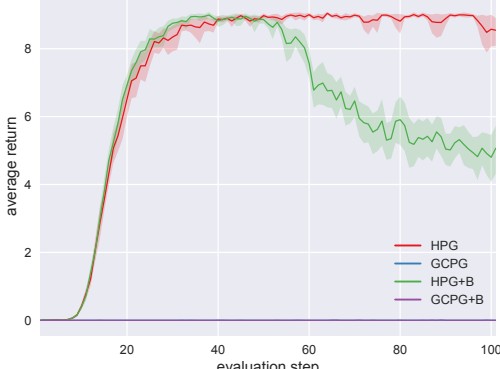

Figure 5: Bit flipping ($k = 16$, batch size 16).

The learning curves for $k = 16$ are presented in Figure 5. Clearly, both GCPG and GCPG+B are unable to obtain policies that perform better than chance, which is explained by the fact that they rarely incorporate reward signals during training. Confirming the importance of hindsight, HPG leads to stable and sample efficient learning. Although HPG+B also obtains excellent policies, they deteriorate upon additional training.

Similar results can be observed for a small batch size (see App. E.3.3). The average performance results documented in Appendix E.3.5 confirm that HPG leads to remarkable sample efficiency. Importantly, Appendices E.3.1 and E.3.2 present hyperparameter sensitivity graphs suggesting that HPG is less sensitive to hyperparameter settings than the other estimators. The same two appendices also document an ablation study where the likelihood ratios are removed from HPG, which notably promotes increased hyperparameter sensitivity. This study confirms the usefulness of the correction prescribed by importance sampling.

## 6.2 GRID WORLD ENVIRONMENTS

In the *grid world* environments that we consider, the agent starts every episode in a (possibly random) position on an $11 \times 11$ grid, and its goal is to reach a randomly chosen (non-initial) position. Some of the positions on the grid may contain impassable obstacles (walls). The actions allow the agent to move in the four cardinal directions. Moving towards walls causes the agent to remain in its current position. A state or goal is represented by a pair of integers between $0$ and $10$. The maximum number of time steps is 32. In the *empty room* environment, the agent starts every episode in the upper left corner of the grid, and there are no walls. In the *four rooms* environment (Sutton et al., 1999b), the agent starts every episode in one of the four corners of the grid (see Fig. 1). There are walls that partition the grid into four rooms, such that each room provides access to two other rooms through single openings (doors). With probability $0.2$, the action chosen by the agent is ignored and replaced by a random action.

Figure 6 shows the learning curves for the empty room environment. Clearly, every estimator obtains excellent policies, although HPG and HPG+B improve sample efficiency by at least 200 batches. The learning curves for the four rooms environment are presented in Figure 7. In this surprisingly challenging environment, every estimator obtains unsatisfactory policies. However, it is still clear that HPG and HPG+B improve sample efficiency. In contrast to the experiments presented in the previous section, HPG+B does not give rise to instability, which we attribute to easier value function estimation. Similar results can be observed for a small batch size (see App. E.3.3). HPG achieves the best average performance in every grid world experiment except for a single case, where the best average performance is achieved by HPG+B (see App. E.3.5). The hyperparameter sensitivity graphs presented in Appendices E.3.1 and E.3.2 once again suggest that HPG is less sensitive to hyperparameter choices, and that ignoring likelihood ratios promotes increased sensitivity (at least in the four rooms environment).

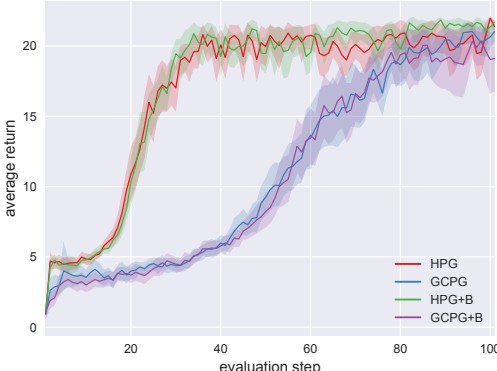

Figure 6: Empty room (batch size 16).

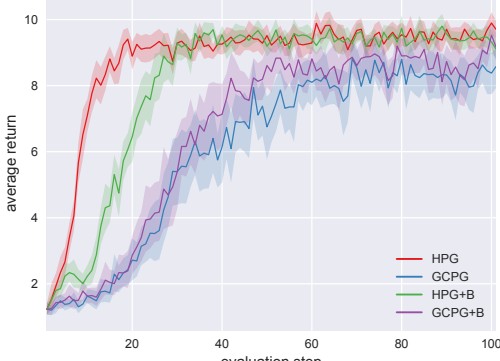

Figure 7: Four rooms (batch size 16).

### 6.3 Ms. Pac-man environment

The *Ms. Pac-man* environment is a variant of the homonymous game for ATARI 2600 (see Fig. 2). The agent starts every episode close to the center of the map, and its goal is to reach a randomly chosen (non-initial) position on a $14 \times 19$ grid defined on the game screen. The actions allow the agent to move in the four cardinal directions for 13 game ticks. A state is represented by the result of preprocessing a sequence of game screens (images) as described in Appendix E.1. A goal is represented by a pair of integers. The maximum number of time steps is 28, although an episode will also end if the agent is captured by an enemy. In comparison to the grid world environments considered in the previous section, this environment is additionally challenging due to its high-dimensional states and the presence of enemies.

Figure 8 presents the learning curves for a medium batch size. Approximate value function baselines are excluded from this experiment due to the significant cost of systematic hyperparameter search. Although HPG obtains better policies during early training, GCPG obtains better final policies. However, for such a medium batch size, only 3 active goals per episode (out of potentially 28) are subsampled for HPG. Although this harsh subsampling brings computational efficiency, it also appears to handicap the estimator. This hypothesis is supported by the fact that HPG outperforms GCPG for a small batch size, when all active goals are used (see Apps. E.3.3 and E.3.5). Policies obtained using each estimator are illustrated by videos included on the project website.

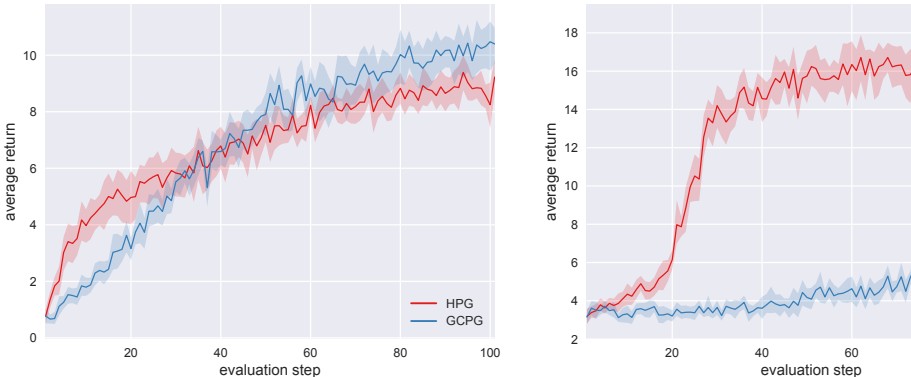

Figure 8: Ms. Pac-man (batch size 16).      Figure 9: FetchPush (batch size 16).

### 6.4 FetchPush environment

The *FetchPush* environment is a variant of the environment recently proposed by Plappert et al. (2018) to assess goal-conditional policy learning algorithms in a challenging task of practical interest (see Fig. 3). In a simulation, a robotic arm with seven degrees of freedom is required to push a randomly placed object (block) towards a randomly chosen position. The arm starts every episode in the same configuration. In contrast to the original environment, the actions in our variant allow increasing the desired velocity of the gripper along each of two orthogonal directions by $\pm 0.1$ or $\pm 1$, leading to a total of eight actions. A state is represented by a 28-dimensional real vector that contains the following information: positions of the gripper and block; rotational and positional velocities of the gripper and block; relative position of the block with respect to the gripper; state of the gripper; and current desired velocity of the gripper along each direction. A goal is represented by three coordinates. The maximum number of time steps is 50.

Figure 9 presents the learning curves for a medium batch size. HPG obtains good policies after a reasonable number of batches, in sharp contrast to GCPG. For such a medium batch size, only 3 active goals per episode (out of potentially 50) are subsampled for HPG, showing that subsampling is a viable alternative to reduce the computational cost of hindsight. Similar results are observed for a small batch size, when all active goals are used (see Apps. E.3.3 and E.3.5). Policies obtained using each estimator are illustrated by videos included on the project website.

# 7   CONCLUSION

We introduced techniques that enable learning goal-conditional policies using hindsight. In this context, hindsight refers to the capacity to exploit information about the degree to which an arbitrary goal has been achieved while another goal was intended. Prior to our work, hindsight has been limited to off-policy reinforcement learning algorithms that rely on experience replay (Andrychowicz et al., 2017) and policy search based on Bayesian optimization (Karkus et al., 2016).

In addition to the fundamental hindsight policy gradient, our technical results include its baseline and advantage formulations. These results are based on a self-contained goal-conditional policy framework that is also introduced in this text. Besides the straightforward estimator built upon the per-decision hindsight policy gradient, we also presented a consistent estimator inspired by weighted importance sampling, together with the corresponding baseline formulation. A variant of this estimator leads to remarkable comparative sample efficiency on a diverse selection of sparse-reward environments, especially in cases where direct reward signals are extremely difficult to obtain. This crucial feature allows natural task formulations that require just trivial reward shaping.

The main drawback of hindsight policy gradient estimators appears to be their computational cost, which is directly related to the number of active goals in a batch. This issue may be mitigated by subsampling active goals, which generally leads to inconsistent estimators. Fortunately, our experiments suggest that this is a viable alternative. Note that the success of hindsight experience replay also depends on an active goal subsampling heuristic (Andrychowicz et al., 2017, Sec. 4.5). The inconsistent hindsight policy gradient estimator with a value function baseline employed in our experiments sometimes leads to unstable learning, which is likely related to the difficulty of fitting such a value function without hindsight. This hypothesis is consistent with the fact that such instability is observed only in the most extreme examples of sparse-reward environments. Although our preliminary experiments in using hindsight to fit a value function baseline have been successful, this may be accomplished in several ways, and requires a careful study of its own. Further experiments are also required to evaluate hindsight on dense-reward environments.

There are many possibilities for future work besides integrating hindsight policy gradients into systems that rely on goal-conditional policies: deriving additional estimators; implementing and evaluating hindsight (advantage) actor-critic methods; assessing whether hindsight policy gradients can successfully circumvent catastrophic forgetting during curriculum learning of goal-conditional policies; approximating the reward function to reduce required supervision; analysing the variance of the proposed estimators; studying the impact of active goal subsampling; and evaluating every technique on continuous action spaces.

## ACKNOWLEDGMENTS

We thank Sjoerd van Steenkiste, Klaus Greff, Imanol Schlag, and the anonymous reviewers for their valuable feedback. This research was supported by the Swiss National Science Foundation (grant 200021_165675/1) and CAPES (Filipe Mutz, PDSE, 88881.133206/2016-01). We are grateful to Nvidia Corporation for donating a *DGX-1* machine and to IBM for donating a *Minsky* machine.

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

## A  GOAL-CONDITIONAL POLICY GRADIENTS

This appendix contains proofs related to the results presented in Section 3: Theorem 3.1 (App. A.2), Theorem 3.2 (App. A.4), and Theorem 3.3 (App. A.6). Appendix A.7 presents optimal constant baselines for goal-conditional policies. The remaining subsections contain auxiliary results.

### A.1  THEOREM A.1

**Theorem A.1.** *The gradient $\nabla\eta(\boldsymbol{\theta})$ of the expected return with respect to $\boldsymbol{\theta}$ is given by*

$$\nabla\eta(\boldsymbol{\theta}) = \sum_g p(g) \sum_{\boldsymbol{\tau}} p(\boldsymbol{\tau} \mid g, \boldsymbol{\theta}) \left[ \sum_{t=1}^{T-1} \nabla \log p(a_t \mid s_t, g, \boldsymbol{\theta}) \right] \left[ \sum_{t=1}^{T} r(s_t, g) \right]. \tag{13}$$

*Proof.* The partial derivative $\partial\eta(\boldsymbol{\theta})/\partial\theta_j$ of the expected return $\eta(\boldsymbol{\theta})$ with respect to $\theta_j$ is given by

$$\frac{\partial}{\partial\theta_j}\eta(\boldsymbol{\theta}) = \sum_g p(g) \sum_{\boldsymbol{\tau}} \frac{\partial}{\partial\theta_j} p(\boldsymbol{\tau} \mid g, \boldsymbol{\theta}) \sum_{t=1}^{T} r(s_t, g). \tag{14}$$

The *likelihood-ratio trick* allows rewriting the previous equation as

$$\frac{\partial}{\partial\theta_j}\eta(\boldsymbol{\theta}) = \sum_g p(g) \sum_{\boldsymbol{\tau}} p(\boldsymbol{\tau} \mid g, \boldsymbol{\theta}) \frac{\partial}{\partial\theta_j} \log p(\boldsymbol{\tau} \mid g, \boldsymbol{\theta}) \sum_{t=1}^{T} r(s_t, g). \tag{15}$$

Note that

$$\log p(\boldsymbol{\tau} \mid g, \boldsymbol{\theta}) = \log p(s_1) + \sum_{t=1}^{T-1} \log p(a_t \mid s_t, g, \boldsymbol{\theta}) + \sum_{t=1}^{T-1} \log p(s_{t+1} \mid s_t, a_t). \tag{16}$$

Therefore,

$$\frac{\partial}{\partial\theta_j}\eta(\boldsymbol{\theta}) = \sum_g p(g) \sum_{\boldsymbol{\tau}} p(\boldsymbol{\tau} \mid g, \boldsymbol{\theta}) \left[ \sum_{t=1}^{T-1} \frac{\partial}{\partial\theta_j} \log p(a_t \mid s_t, g, \boldsymbol{\theta}) \right] \left[ \sum_{t=1}^{T} r(s_t, g) \right]. \tag{17}$$

$\square$

### A.2  THEOREM 3.1

**Theorem 3.1** (Goal-conditional policy gradient). *The gradient $\nabla\eta(\boldsymbol{\theta})$ of the expected return with respect to $\boldsymbol{\theta}$ is given by*

$$\nabla\eta(\boldsymbol{\theta}) = \sum_g p(g) \sum_{\boldsymbol{\tau}} p(\boldsymbol{\tau} \mid g, \boldsymbol{\theta}) \sum_{t=1}^{T-1} \nabla \log p(a_t \mid s_t, g, \boldsymbol{\theta}) \sum_{t'=t+1}^{T} r(s_{t'}, g). \tag{3}$$

*Proof.* Starting from Eq. 17, the partial derivative $\partial\eta(\boldsymbol{\theta})/\partial\theta_j$ of $\eta(\boldsymbol{\theta})$ with respect to $\theta_j$ is given by

$$\frac{\partial}{\partial\theta_j}\eta(\boldsymbol{\theta}) = \sum_g p(g \mid \boldsymbol{\theta}) \sum_{\boldsymbol{\tau}} p(\boldsymbol{\tau} \mid g, \boldsymbol{\theta}) \sum_{t=1}^{T} r(s_t, g) \sum_{t'=1}^{T-1} \frac{\partial}{\partial\theta_j} \log p(a_{t'} \mid s_{t'}, g, \boldsymbol{\theta}). \tag{18}$$

The previous equation can be rewritten as

$$\frac{\partial}{\partial\theta_j}\eta(\boldsymbol{\theta}) = \sum_{t=1}^{T} \sum_{t'=1}^{T-1} \mathbb{E}\left[ r(S_t, G) \frac{\partial}{\partial\theta_j} \log p(A_{t'} \mid S_{t'}, G, \boldsymbol{\theta}) \mid \boldsymbol{\theta} \right]. \tag{19}$$

Let $c$ denote an expectation inside Eq. 19 for $t' \geq t$. In that case, $A_{t'} \perp\!\!\!\perp S_t \mid S_{t'}, G, \boldsymbol{\Theta}$, and so

$$c = \sum_{s_t} \sum_{s_{t'}} \sum_{g} \sum_{a_{t'}} p(a_{t'} \mid s_{t'}, g, \boldsymbol{\theta}) p(s_t, s_{t'}, g \mid \boldsymbol{\theta}) r(s_t, g) \frac{\partial}{\partial \theta_j} \log p(a_{t'} \mid s_{t'}, g, \boldsymbol{\theta}). \quad (20)$$

Reversing the likelihood-ratio trick,

$$c = \sum_{s_t} \sum_{s_{t'}} \sum_{g} p(s_t, s_{t'}, g \mid \boldsymbol{\theta}) r(s_t, g) \frac{\partial}{\partial \theta_j} \sum_{a_{t'}} p(a_{t'} \mid s_{t'}, g, \boldsymbol{\theta}) = 0. \quad (21)$$

Therefore, the terms where $t' \geq t$ can be dismissed from Eq. 19, leading to

$$\frac{\partial}{\partial \theta_j} \eta(\boldsymbol{\theta}) = \mathbb{E}\left[ \sum_{t=1}^{T} r(S_t, G) \sum_{t'=1}^{t-1} \frac{\partial}{\partial \theta_j} \log p(A_{t'} \mid S_{t'}, G, \boldsymbol{\theta}) \mid \boldsymbol{\theta} \right]. \quad (22)$$

The previous equation can be conveniently rewritten as

$$\frac{\partial}{\partial \theta_j} \eta(\boldsymbol{\theta}) = \mathbb{E}\left[ \sum_{t=1}^{T-1} \frac{\partial}{\partial \theta_j} \log p(A_t \mid S_t, G, \boldsymbol{\theta}) \sum_{t'=t+1}^{T} r(S_{t'}, G) \mid \boldsymbol{\theta} \right]. \quad (23)$$

$\square$

## A.3 Lemma A.1

**Lemma A.1.** *For every $j, t, \boldsymbol{\theta}$, and associated real-valued (baseline) function $b_t^{\boldsymbol{\theta}}$,*

$$\sum_{t=1}^{T-1} \mathbb{E}\left[ \frac{\partial}{\partial \theta_j} \log p(A_t \mid S_t, G, \boldsymbol{\theta}) b_t^{\boldsymbol{\theta}}(S_t, G) \mid \boldsymbol{\theta} \right] = 0. \quad (24)$$

*Proof.* Letting $c$ denote an expectation inside Eq. 24,

$$c = \sum_{s_t} \sum_{g} \sum_{a_t} p(a_t \mid s_t, g, \boldsymbol{\theta}) p(s_t, g \mid \boldsymbol{\theta}) \frac{\partial}{\partial \theta_j} \log p(a_t \mid s_t, g, \boldsymbol{\theta}) b_t^{\boldsymbol{\theta}}(s_t, g). \quad (25)$$

Reversing the likelihood-ratio trick,

$$c = \sum_{s_t} \sum_{g} p(s_t, g \mid \boldsymbol{\theta}) b_t^{\boldsymbol{\theta}}(s_t, g) \frac{\partial}{\partial \theta_j} \sum_{a_t} p(a_t \mid s_t, g, \boldsymbol{\theta}) = 0. \quad (26)$$

$\square$

## A.4 Theorem 3.2

**Theorem 3.2** (Goal-conditional policy gradient, baseline formulation). *For every $t, \boldsymbol{\theta}$, and associated real-valued (baseline) function $b_t^{\boldsymbol{\theta}}$, the gradient $\nabla \eta(\boldsymbol{\theta})$ of the expected return with respect to $\boldsymbol{\theta}$ is given by*

$$\nabla \eta(\boldsymbol{\theta}) = \sum_{g} p(g) \sum_{\boldsymbol{\tau}} p(\boldsymbol{\tau} \mid g, \boldsymbol{\theta}) \sum_{t=1}^{T-1} \nabla \log p(a_t \mid s_t, g, \boldsymbol{\theta}) \left[ \left[ \sum_{t'=t+1}^{T} r(s_{t'}, g) \right] - b_t^{\boldsymbol{\theta}}(s_t, g) \right]. \quad (4)$$

*Proof.* The result is obtained by subtracting Eq. 24 from Eq. 23. Importantly, for every combination of $\boldsymbol{\theta}$ and $t$, it would also be possible to have a distinct baseline function for each parameter in $\boldsymbol{\theta}$. $\square$

## A.5 LEMMA A.2

**Lemma A.2.** *The gradient $\nabla\eta(\boldsymbol{\theta})$ of the expected return with respect to $\boldsymbol{\theta}$ is given by*

$$\nabla\eta(\boldsymbol{\theta}) = \sum_g p(g) \sum_{\boldsymbol{\tau}} p(\boldsymbol{\tau} \mid g, \boldsymbol{\theta}) \sum_{t=1}^{T-1} \nabla \log p(a_t \mid s_t, g, \boldsymbol{\theta}) Q_t^{\boldsymbol{\theta}}(s_t, a_t, g). \tag{27}$$

*Proof.* Starting from Eq. 23 and rearranging terms,

$$\frac{\partial}{\partial \theta_j} \eta(\boldsymbol{\theta}) = \sum_{t=1}^{T-1} \sum_g \sum_{s_t} \sum_{a_t} p(s_t, a_t, g \mid \boldsymbol{\theta}) \frac{\partial}{\partial \theta_j} \log p(a_t \mid s_t, g, \boldsymbol{\theta}) \sum_{s_{t+1:T}} p(s_{t+1:T} \mid s_t, a_t, g, \boldsymbol{\theta}) \sum_{t'=t+1}^{T} r(s_{t'}, g). \tag{28}$$

By the definition of action-value function,

$$\frac{\partial}{\partial \theta_j} \eta(\boldsymbol{\theta}) = \mathbb{E}\left[ \sum_{t=1}^{T-1} \frac{\partial}{\partial \theta_j} \log p(A_t \mid S_t, G, \boldsymbol{\theta}) Q_t^{\boldsymbol{\theta}}(S_t, A_t, G) \mid \boldsymbol{\theta} \right]. \tag{29}$$

$\square$

## A.6 THEOREM 3.3

**Theorem 3.3** (Goal-conditional policy gradient, advantage formulation). *The gradient $\nabla\eta(\boldsymbol{\theta})$ of the expected return with respect to $\boldsymbol{\theta}$ is given by*

$$\nabla\eta(\boldsymbol{\theta}) = \sum_g p(g) \sum_{\boldsymbol{\tau}} p(\boldsymbol{\tau} \mid g, \boldsymbol{\theta}) \sum_{t=1}^{T-1} \nabla \log p(a_t \mid s_t, g, \boldsymbol{\theta}) A_t^{\boldsymbol{\theta}}(s_t, a_t, g). \tag{5}$$

*Proof.* The result is obtained by choosing $b_t^{\boldsymbol{\theta}} = V_t^{\boldsymbol{\theta}}$ and subtracting Eq. 24 from Eq. 29. $\square$

## A.7 THEOREM A.2

For arbitrary $j$ and $\boldsymbol{\theta}$, consider the following definitions of $f$ and $h$.

$$f(\boldsymbol{\tau}, g) = \sum_{t=1}^{T-1} \frac{\partial}{\partial \theta_j} \log p(a_t \mid s_t, g, \boldsymbol{\theta}) \sum_{t'=t+1}^{T} r(s_{t'}, g), \tag{30}$$

$$h(\boldsymbol{\tau}, g) = \sum_{t=1}^{T-1} \frac{\partial}{\partial \theta_j} \log p(a_t \mid s_t, g, \boldsymbol{\theta}). \tag{31}$$

For every $b_j \in \mathbb{R}$, using Theorem 3.1 and the fact that $\mathbb{E}\left[h(\boldsymbol{\mathcal{T}}, G) \mid \boldsymbol{\theta}\right] = 0$ by Lemma A.1,

$$\frac{\partial}{\partial \theta_j} \eta(\boldsymbol{\theta}) = \mathbb{E}\left[f(\boldsymbol{\mathcal{T}}, G) \mid \boldsymbol{\theta}\right] = \mathbb{E}\left[f(\boldsymbol{\mathcal{T}}, G) - b_j h(\boldsymbol{\mathcal{T}}, G) \mid \boldsymbol{\theta}\right]. \tag{32}$$

**Theorem A.2.** *Assuming $\mathrm{Var}\left[h(\boldsymbol{\mathcal{T}}, G) \mid \boldsymbol{\theta}\right] > 0$, the (optimal constant baseline) $b_j$ that minimizes $\mathrm{Var}\left[f(\boldsymbol{\mathcal{T}}, G) - b_j h(\boldsymbol{\mathcal{T}}, G) \mid \boldsymbol{\theta}\right]$ is given by*

$$b_j = \frac{\mathbb{E}\left[f(\boldsymbol{\mathcal{T}}, G) h(\boldsymbol{\mathcal{T}}, G) \mid \boldsymbol{\theta}\right]}{\mathbb{E}\left[h(\boldsymbol{\mathcal{T}}, G)^2 \mid \boldsymbol{\theta}\right]}. \tag{33}$$

*Proof.* The result is an application of Lemma D.4. $\square$

## B  HINDSIGHT POLICY GRADIENTS

This appendix contains proofs related to the results presented in Section 4: Theorem 4.1 (App. B.1), Theorem 4.2 (App. B.2), Lemma 4.1 (App. B.3), Theorem B.1 (App. B.4), and Theorem 4.3 (App. B.6). Appendix B.7 presents optimal constant baselines for hindsight policy gradients. Appendix B.5 contains an auxiliary result.

### B.1  THEOREM 4.1

The following theorem relies on importance sampling, a traditional technique used to obtain estimates related to a random variable $X \sim p$ using samples from an arbitrary positive distribution $q$. This technique relies on the following equalities:

$$\mathbb{E}_{p(X)}[f(X)] = \sum_x p(x)f(x) = \sum_x \frac{q(x)}{q(x)}p(x)f(x) = \mathbb{E}_{q(X)}\left[\frac{p(X)}{q(X)}f(X)\right]. \quad (34)$$

**Theorem 4.1** (Every-decision hindsight policy gradient). *For an arbitrary (original) goal $g'$, the gradient $\nabla\eta(\boldsymbol{\theta})$ of the expected return with respect to $\boldsymbol{\theta}$ is given by*

$$\nabla\eta(\boldsymbol{\theta}) = \sum_{\boldsymbol{\tau}} p(\boldsymbol{\tau} \mid g', \boldsymbol{\theta}) \sum_g p(g) \sum_{t=1}^{T-1} \nabla \log p(a_t \mid s_t, g, \boldsymbol{\theta}) \sum_{t'=t+1}^{T} \left[\prod_{k=1}^{T-1} \frac{p(a_k \mid s_k, g, \boldsymbol{\theta})}{p(a_k \mid s_k, g', \boldsymbol{\theta})}\right] r(s_{t'}, g). \quad (6)$$

*Proof.* Starting from Theorem 3.1, importance sampling allows rewriting the partial derivative $\partial\eta(\boldsymbol{\theta})/\partial\theta_j$ as

$$\frac{\partial}{\partial\theta_j}\eta(\boldsymbol{\theta}) = \sum_g p(g) \sum_{\boldsymbol{\tau}} \frac{p(\boldsymbol{\tau} \mid g', \boldsymbol{\theta})}{p(\boldsymbol{\tau} \mid g', \boldsymbol{\theta})} p(\boldsymbol{\tau} \mid g, \boldsymbol{\theta}) \sum_{t=1}^{T-1} \frac{\partial}{\partial\theta_j} \log p(a_t \mid s_t, g, \boldsymbol{\theta}) \sum_{t'=t+1}^{T} r(s_{t'}, g). \quad (35)$$

Using Equation 1,

$$\frac{\partial}{\partial\theta_j}\eta(\boldsymbol{\theta}) = \sum_g p(g) \sum_{\boldsymbol{\tau}} p(\boldsymbol{\tau} \mid g', \boldsymbol{\theta}) \left[\prod_{k=1}^{T-1} \frac{p(a_k \mid s_k, g, \boldsymbol{\theta})}{p(a_k \mid s_k, g', \boldsymbol{\theta})}\right] \sum_{t=1}^{T-1} \frac{\partial}{\partial\theta_j} \log p(a_t \mid s_t, g, \boldsymbol{\theta}) \sum_{t'=t+1}^{T} r(s_{t'}, g).$$
$$(36)$$

$\square$

### B.2  THEOREM 4.2

**Theorem 4.2** (Per-decision hindsight policy gradient). *For an arbitrary (original) goal $g'$, the gradient $\nabla\eta(\boldsymbol{\theta})$ of the expected return with respect to $\boldsymbol{\theta}$ is given by*

$$\nabla\eta(\boldsymbol{\theta}) = \sum_{\boldsymbol{\tau}} p(\boldsymbol{\tau} \mid g', \boldsymbol{\theta}) \sum_g p(g) \sum_{t=1}^{T-1} \nabla \log p(a_t \mid s_t, g, \boldsymbol{\theta}) \sum_{t'=t+1}^{T} \left[\prod_{k=1}^{t'-1} \frac{p(a_k \mid s_k, g, \boldsymbol{\theta})}{p(a_k \mid s_k, g', \boldsymbol{\theta})}\right] r(s_{t'}, g). \quad (7)$$

*Proof.* Starting from Eq. 36, the partial derivative $\partial\eta(\boldsymbol{\theta})/\partial\theta_j$ can be rewritten as

$$\frac{\partial}{\partial\theta_j}\eta(\boldsymbol{\theta}) = \sum_g p(g) \sum_{t=1}^{T-1} \sum_{t'=t+1}^{T} \sum_{\boldsymbol{\tau}} p(\boldsymbol{\tau} \mid g', \boldsymbol{\theta}) \left[\prod_{k=1}^{T-1} \frac{p(a_k \mid s_k, g, \boldsymbol{\theta})}{p(a_k \mid s_k, g', \boldsymbol{\theta})}\right] \frac{\partial}{\partial\theta_j} \log p(a_t \mid s_t, g, \boldsymbol{\theta}) r(s_{t'}, g). \quad (37)$$

If we split every trajectory into states and actions before and after $t'$, then $\partial\eta(\boldsymbol{\theta})/\partial\theta_j$ is given by

$$\sum_g p(g) \sum_{t=1}^{T-1} \sum_{t'=t+1}^{T} \sum_{s_{1:t'-1}} \sum_{a_{1:t'-1}} p(s_{1:t'-1}, a_{1:t'-1} \mid g', \boldsymbol{\theta}) \left[\prod_{k=1}^{t'-1} \frac{p(a_k \mid s_k, g, \boldsymbol{\theta})}{p(a_k \mid s_k, g', \boldsymbol{\theta})}\right] \frac{\partial}{\partial\theta_j} \log p(a_t \mid s_t, g, \boldsymbol{\theta})z, \quad (38)$$

where $z$ is defined by

$$z = \sum_{s_{t':T}} \sum_{a_{t':T-1}} p(s_{t':T}, a_{t':T-1} \mid s_{1:t'-1}, a_{1:t'-1}, g', \boldsymbol{\theta}) \left[ \prod_{k=t'}^{T-1} \frac{p(a_k \mid s_k, g, \boldsymbol{\theta})}{p(a_k \mid s_k, g', \boldsymbol{\theta})} \right] r(s_{t'}, g). \quad (39)$$

Using Lemma D.2 and canceling terms,

$$z = \sum_{s_{t':T}} \sum_{a_{t':T-1}} p(s_{t'} \mid s_{t'-1}, a_{t'-1}) \left[ \prod_{k=t'}^{T-1} p(a_k \mid s_k, g, \boldsymbol{\theta}) p(s_{k+1} \mid s_k, a_k) \right] r(s_{t'}, g). \quad (40)$$

Using Lemma D.2 once again,

$$z = \sum_{s_{t':T}} \sum_{a_{t':T-1}} p(s_{t':T}, a_{t':T-1} \mid s_{1:t'-1}, a_{1:t'-1}, g, \boldsymbol{\theta}) r(s_{t'}, g). \quad (41)$$

Using the fact that $S_{t'} \perp\!\!\!\perp G \mid S_{1:t'-1}, A_{1:t'-1}, \boldsymbol{\Theta}$,

$$z = \sum_{s_{t'}} r(s_{t'}, g) p(s_{t'} \mid s_{1:t'-1}, a_{1:t'-1}, g, \boldsymbol{\theta}) = \sum_{s_{t'}} r(s_{t'}, g) p(s_{t'} \mid s_{1:t'-1}, a_{1:t'-1}, g', \boldsymbol{\theta}). \quad (42)$$

Substituting $z$ into Expression 38 and returning to an expectation over trajectories,

$$\frac{\partial}{\partial \theta_j} \eta(\boldsymbol{\theta}) = \sum_{\boldsymbol{\tau}} p(\boldsymbol{\tau} \mid g', \boldsymbol{\theta}) \sum_g p(g) \sum_{t=1}^{T-1} \frac{\partial}{\partial \theta_j} \log p(a_t \mid s_t, g, \boldsymbol{\theta}) \sum_{t'=t+1}^{T} \left[ \prod_{k=1}^{t'-1} \frac{p(a_k \mid s_k, g, \boldsymbol{\theta})}{p(a_k \mid s_k, g', \boldsymbol{\theta})} \right] r(s_{t'}, g). \quad (43)$$

$\square$

## B.3 LEMMA 4.1

**Lemma 4.1.** *For every $g'$, $t$, $\boldsymbol{\theta}$, and associated real-valued (baseline) function $b_t^{\boldsymbol{\theta}}$,*

$$\sum_{\boldsymbol{\tau}} p(\boldsymbol{\tau} \mid g', \boldsymbol{\theta}) \sum_g p(g) \sum_{t=1}^{T-1} \nabla \log p(a_t \mid s_t, g, \boldsymbol{\theta}) \left[ \prod_{k=1}^{t} \frac{p(a_k \mid s_k, g, \boldsymbol{\theta})}{p(a_k \mid s_k, g', \boldsymbol{\theta})} \right] b_t^{\boldsymbol{\theta}}(s_t, g) = \mathbf{0}. \quad (8)$$

*Proof.* Let $c$ denote the $j$-th element of the vector in the left-hand side of Eq. 8, such that

$$c = \sum_g p(g) \sum_{t=1}^{T-1} \mathbb{E}\left[ \frac{\partial}{\partial \theta_j} \log p(A_t \mid S_t, g, \boldsymbol{\theta}) \left[ \prod_{k=1}^{t} \frac{p(A_k \mid S_k, g, \boldsymbol{\theta})}{p(A_k \mid S_k, g', \boldsymbol{\theta})} \right] b_t^{\boldsymbol{\theta}}(S_t, g) \mid g', \boldsymbol{\theta} \right]. \quad (44)$$

Using Lemma D.1 and writing the expectations explicitly,

$$c = \sum_g p(g) \sum_{t=1}^{T-1} \sum_{s_{1:t}} \sum_{a_{1:t}} p(s_{1:t}, a_{1:t} \mid g', \boldsymbol{\theta}) \frac{\partial}{\partial \theta_j} \log p(a_t \mid s_t, g, \boldsymbol{\theta}) \frac{p(s_{1:t}, a_{1:t} \mid g, \boldsymbol{\theta})}{p(s_{1:t}, a_{1:t} \mid g', \boldsymbol{\theta})} b_t^{\boldsymbol{\theta}}(s_t, g). \quad (45)$$

Canceling terms, using Lemma D.1 once again, and reversing the likelihood-ratio trick,

$$c = \sum_g p(g) \sum_{t=1}^{T-1} \sum_{s_{1:t}} \sum_{a_{1:t}} \frac{\partial}{\partial \theta_j} p(a_t \mid s_t, g, \boldsymbol{\theta}) \left[ p(s_1) \prod_{k=1}^{t-1} p(a_k \mid s_k, g, \boldsymbol{\theta}) p(s_{k+1} \mid s_k, a_k) \right] b_t^{\boldsymbol{\theta}}(s_t, g). \quad (46)$$

Pushing constants outside the summation over actions at time step $t$,

$$c = \sum_g p(g) \sum_{t=1}^{T-1} \sum_{s_{1:t}} \sum_{a_{1:t-1}} \left[ p(s_1) \prod_{k=1}^{t-1} p(a_k \mid s_k, g, \boldsymbol{\theta}) p(s_{k+1} \mid s_k, a_k) \right] b_t^{\boldsymbol{\theta}}(s_t, g) \frac{\partial}{\partial \theta_j} \sum_{a_t} p(a_t \mid s_t, g, \boldsymbol{\theta}) = 0. \quad (47)$$

$\square$

### B.4 THEOREM B.1

**Theorem B.1** (Hindsight policy gradient, baseline formulation). *For every $g'$, $t$, $\boldsymbol{\theta}$, and associated real-valued (baseline) function $b_t^{\boldsymbol{\theta}}$, the gradient $\nabla \eta(\boldsymbol{\theta})$ of the expected return with respect to $\boldsymbol{\theta}$ is given by*

$$\nabla \eta(\boldsymbol{\theta}) = \sum_{\boldsymbol{\tau}} p(\boldsymbol{\tau} \mid g', \boldsymbol{\theta}) \sum_g p(g) \sum_{t=1}^{T-1} \nabla \log p(a_t \mid s_t, g, \boldsymbol{\theta}) z, \tag{48}$$

*where*

$$z = \left[ \sum_{t'=t+1}^{T} \left[ \prod_{k=1}^{t'-1} \frac{p(a_k \mid s_k, g, \boldsymbol{\theta})}{p(a_k \mid s_k, g', \boldsymbol{\theta})} \right] r(s_{t'}, g) \right] - \left[ \prod_{k=1}^{t} \frac{p(a_k \mid s_k, g, \boldsymbol{\theta})}{p(a_k \mid s_k, g', \boldsymbol{\theta})} \right] b_t^{\boldsymbol{\theta}}(s_t, g). \tag{49}$$

*Proof.* The result is obtained by subtracting Eq. 8 from Eq. 7. Importantly, for every combination of $\boldsymbol{\theta}$ and $t$, it would also be possible to have a distinct baseline function for each parameter in $\boldsymbol{\theta}$. $\square$

### B.5 LEMMA B.1

**Lemma B.1** (Hindsight policy gradient, action-value formulation). *For an arbitrary goal $g'$, the gradient $\nabla \eta(\boldsymbol{\theta})$ of the expected return with respect to $\boldsymbol{\theta}$ is given by*

$$\nabla \eta(\boldsymbol{\theta}) = \sum_{\boldsymbol{\tau}} p(\boldsymbol{\tau} \mid g', \boldsymbol{\theta}) \sum_g p(g) \sum_{t=1}^{T-1} \nabla \log p(a_t \mid s_t, g, \boldsymbol{\theta}) \left[ \prod_{k=1}^{t} \frac{p(a_k \mid s_k, g, \boldsymbol{\theta})}{p(a_k \mid s_k, g', \boldsymbol{\theta})} \right] Q_t^{\boldsymbol{\theta}}(s_t, a_t, g). \tag{50}$$

*Proof.* Starting from Eq. 29, the partial derivative $\partial \eta(\boldsymbol{\theta}) / \partial \theta_j$ can be written as

$$\frac{\partial}{\partial \theta_j} \eta(\boldsymbol{\theta}) = \sum_{t=1}^{T-1} \sum_g p(g) \sum_{s_{1:t}} \sum_{a_{1:t}} p(s_{1:t}, a_{1:t} \mid g, \boldsymbol{\theta}) \frac{\partial}{\partial \theta_j} \log p(a_t \mid s_t, g, \boldsymbol{\theta}) Q_t^{\boldsymbol{\theta}}(s_t, a_t, g). \tag{51}$$

Using importance sampling, for an arbitrary goal $g'$,

$$\frac{\partial}{\partial \theta_j} \eta(\boldsymbol{\theta}) = \sum_g p(g) \sum_{t=1}^{T-1} \sum_{s_{1:t}} \sum_{a_{1:t}} p(s_{1:t}, a_{1:t} \mid g', \boldsymbol{\theta}) \frac{p(s_{1:t}, a_{1:t} \mid g, \boldsymbol{\theta})}{p(s_{1:t}, a_{1:t} \mid g', \boldsymbol{\theta})} \frac{\partial}{\partial \theta_j} \log p(a_t \mid s_t, g, \boldsymbol{\theta}) Q_t^{\boldsymbol{\theta}}(s_t, a_t, g). \tag{52}$$

Using Lemma D.1 and rewriting the previous equation using expectations,

$$\frac{\partial}{\partial \theta_j} \eta(\boldsymbol{\theta}) = \sum_g p(g) \mathbb{E} \left[ \sum_{t=1}^{T-1} \frac{\partial}{\partial \theta_j} \log p(A_t \mid S_t, g, \boldsymbol{\theta}) \left[ \prod_{k=1}^{t} \frac{p(A_k \mid S_k, g, \boldsymbol{\theta})}{p(A_k \mid S_k, g', \boldsymbol{\theta})} \right] Q_t^{\boldsymbol{\theta}}(S_t, A_t, g) \mid g', \boldsymbol{\theta} \right]. \tag{53}$$

$\square$

### B.6 THEOREM 4.3

**Theorem 4.3** (Hindsight policy gradient, advantage formulation). *For an arbitrary (original) goal $g'$, the gradient $\nabla \eta(\boldsymbol{\theta})$ of the expected return with respect to $\boldsymbol{\theta}$ is given by*

$$\nabla \eta(\boldsymbol{\theta}) = \sum_{\boldsymbol{\tau}} p(\boldsymbol{\tau} \mid g', \boldsymbol{\theta}) \sum_g p(g) \sum_{t=1}^{T-1} \nabla \log p(a_t \mid s_t, g, \boldsymbol{\theta}) \left[ \prod_{k=1}^{t} \frac{p(a_k \mid s_k, g, \boldsymbol{\theta})}{p(a_k \mid s_k, g', \boldsymbol{\theta})} \right] A_t^{\boldsymbol{\theta}}(s_t, a_t, g). \tag{9}$$

*Proof.* The result is obtained by choosing $b_t^{\boldsymbol{\theta}} = V_t^{\boldsymbol{\theta}}$ and subtracting Eq. 44 from Eq. 53. $\square$

## B.7   THEOREM B.2

For arbitrary $g', j$, and $\boldsymbol{\theta}$, consider the following definitions of $f$ and $h$.

$$f(\boldsymbol{\tau}) = \sum_g p(g) \sum_{t=1}^{T-1} \frac{\partial}{\partial \theta_j} \log p(a_t \mid s_t, g, \boldsymbol{\theta}) \sum_{t'=t+1}^{T} \left[ \prod_{k=1}^{t'-1} \frac{p(a_k \mid s_k, g, \boldsymbol{\theta})}{p(a_k \mid s_k, g', \boldsymbol{\theta})} \right] r(s_{t'}, g), \qquad (54)$$

$$h(\boldsymbol{\tau}) = \sum_g p(g) \sum_{t=1}^{T-1} \frac{\partial}{\partial \theta_j} \log p(a_t \mid s_t, g, \boldsymbol{\theta}) \prod_{k=1}^{t} \frac{p(a_k \mid s_k, g, \boldsymbol{\theta})}{p(a_k \mid s_k, g', \boldsymbol{\theta})}. \qquad (55)$$

For every $b_j \in \mathbb{R}$, using Theorem 4.2 and the fact that $\mathbb{E}\left[h(\boldsymbol{\mathcal{T}}) \mid g', \boldsymbol{\theta}\right] = 0$ by Lemma 4.1,

$$\frac{\partial}{\partial \theta_j} \eta(\boldsymbol{\theta}) = \mathbb{E}\left[f(\boldsymbol{\mathcal{T}}) \mid g', \boldsymbol{\theta}\right] = \mathbb{E}\left[f(\boldsymbol{\mathcal{T}}) - b_j h(\boldsymbol{\mathcal{T}}) \mid g', \boldsymbol{\theta}\right]. \qquad (56)$$

**Theorem B.2.** *Assuming* $\mathrm{Var}\left[h(\boldsymbol{\mathcal{T}}) \mid g', \boldsymbol{\theta}\right] > 0$, *the (optimal constant baseline)* $b_j$ *that minimizes* $\mathrm{Var}\left[f(\boldsymbol{\mathcal{T}}) - b_j h(\boldsymbol{\mathcal{T}}) \mid g', \boldsymbol{\theta}\right]$ *is given by*

$$b_j = \frac{\mathbb{E}\left[f(\boldsymbol{\mathcal{T}}) h(\boldsymbol{\mathcal{T}}) \mid g', \boldsymbol{\theta}\right]}{\mathbb{E}\left[h(\boldsymbol{\mathcal{T}})^2 \mid g', \boldsymbol{\theta}\right]}. \qquad (57)$$

*Proof.* The result is an application of Lemma D.4. $\qquad\square$

## C HINDSIGHT GRADIENT ESTIMATORS

This appendix contains proofs related to the estimators presented in Section 5: Theorem 5.1 (App. C.1) and Theorem 5.2 (App. C.2). Appendix C.3 presents a result that enables a consistency-preserving *weighted baseline*.

In this appendix, we will consider a dataset $\mathcal{D} = \{(\boldsymbol{\tau}^{(i)}, g^{(i)})\}_{i=1}^{N}$ where each trajectory $\boldsymbol{\tau}^{(i)}$ is obtained using a policy parameterized by $\boldsymbol{\theta}$ in an attempt to achieve a goal $g^{(i)}$ chosen by the environment. Because $\mathcal{D}$ is an *iid* dataset given $\boldsymbol{\Theta}$,

$$p(\mathcal{D} \mid \boldsymbol{\theta}) = p(\boldsymbol{\tau}^{(1:N)}, g^{(1:N)} \mid \boldsymbol{\theta}) = \prod_{i=1}^{N} p(\boldsymbol{\tau}^{(i)}, g^{(i)} \mid \boldsymbol{\theta}) = \prod_{i=1}^{N} p(g^{(i)}) p(\boldsymbol{\tau}^{(i)} \mid g^{(i)}, \boldsymbol{\theta}). \quad (58)$$

### C.1 THEOREM 5.1

**Theorem 5.1.** *The per-decision hindsight policy gradient estimator, given by*

$$\frac{1}{N} \sum_{i=1}^{N} \sum_{g} p(g) \sum_{t=1}^{T-1} \nabla \log p(A_t^{(i)} \mid S_t^{(i)}, G^{(i)} = g, \boldsymbol{\theta}) \sum_{t'=t+1}^{T} \left[ \prod_{k=1}^{t'-1} \frac{p(A_k^{(i)} \mid S_k^{(i)}, G^{(i)} = g, \boldsymbol{\theta})}{p(A_k^{(i)} \mid S_k^{(i)}, G^{(i)}, \boldsymbol{\theta})} \right] r(S_{t'}^{(i)}, g), \quad (11)$$

*is a consistent and unbiased estimator of the gradient $\nabla \eta(\boldsymbol{\theta})$ of the expected return.*

*Proof.* Let $I_j^{(N)}$ denote the $j$-th element of the estimator, which can be written as

$$I_j^{(N)} = \frac{1}{N} \sum_{i=1}^{N} I(\boldsymbol{\mathcal{T}}^{(i)}, G^{(i)}, \boldsymbol{\theta})_j, \quad (59)$$

where

$$I(\boldsymbol{\tau}, g', \boldsymbol{\theta})_j = \sum_{g} p(g) \sum_{t=1}^{T-1} \frac{\partial}{\partial \theta_j} \log p(a_t \mid s_t, g, \boldsymbol{\theta}) \sum_{t'=t+1}^{T} \left[ \prod_{k=1}^{t'-1} \frac{p(a_k \mid s_k, g, \boldsymbol{\theta})}{p(a_k \mid s_k, g', \boldsymbol{\theta})} \right] r(s_{t'}, g). \quad (60)$$

Using Theorem 4.2, the expected value $\mathbb{E}\left[ I_j^{(N)} \mid \boldsymbol{\theta} \right]$ is given by

$$\mathbb{E}\left[ I_j^{(N)} \mid \boldsymbol{\theta} \right] = \frac{1}{N} \sum_{i=1}^{N} \sum_{g^{(i)}} p(g^{(i)}) \mathbb{E}\left[ I(\boldsymbol{\mathcal{T}}^{(i)}, g^{(i)}, \boldsymbol{\theta})_j \mid g^{(i)}, \boldsymbol{\theta} \right] = \frac{1}{N} \sum_{i=1}^{N} \sum_{g^{(i)}} p(g^{(i)}) \frac{\partial}{\partial \theta_j} \eta(\boldsymbol{\theta}) = \frac{\partial}{\partial \theta_j} \eta(\boldsymbol{\theta}). \quad (61)$$

Therefore, $I_j^{(N)}$ is an unbiased estimator of $\partial \eta(\boldsymbol{\theta})/\partial \theta_j$.

Conditionally on $\boldsymbol{\Theta}$, the random variable $I_j^{(N)}$ is an average of iid random variables with expected value $\partial \eta(\boldsymbol{\theta})/\partial \theta_j$ (see Eq. 61). By the strong law of large numbers (Sen & Singer, 1994, Theorem 2.3.13),

$$I_j^{(N)} \xrightarrow{\text{a.s.}} \frac{\partial}{\partial \theta_j} \eta(\boldsymbol{\theta}). \quad (62)$$

Therefore, $I_j^{(N)}$ is a consistent estimator of $\partial \eta(\boldsymbol{\theta})/\partial \theta_j$.

$\square$

### C.2 THEOREM 5.2

**Theorem 5.2.** *The weighted per-decision hindsight policy gradient estimator, given by*

$$\sum_{i=1}^{N} \sum_{g} p(g) \sum_{t=1}^{T-1} \nabla \log p(A_t^{(i)} \mid S_t^{(i)}, G^{(i)} = g, \boldsymbol{\theta}) \sum_{t'=t+1}^{T} \frac{\left[ \prod_{k=1}^{t'-1} \frac{p(A_k^{(i)} \mid S_k^{(i)}, G^{(i)} = g, \boldsymbol{\theta})}{p(A_k^{(i)} \mid S_k^{(i)}, G^{(i)}, \boldsymbol{\theta})} \right] r(S_{t'}^{(i)}, g)}{\sum_{j=1}^{N} \left[ \prod_{k=1}^{t'-1} \frac{p(A_k^{(j)} \mid S_k^{(j)}, G^{(j)} = g, \boldsymbol{\theta})}{p(A_k^{(j)} \mid S_k^{(j)}, G^{(j)}, \boldsymbol{\theta})} \right]}, \quad (12)$$

*is a consistent estimator of the gradient $\nabla \eta(\boldsymbol{\theta})$ of the expected return.*

*Proof.* Let $W_j^{(N)}$ denote the $j$-th element of the estimator, which can be written as

$$W_j^{(N)} = \sum_g p(g) \sum_{t=1}^{T-1} \sum_{t'=t+1}^{T} \frac{X(g,t,t')_j^{(N)}}{Y(g,t,t')_j^{(N)}}, \tag{63}$$

where

$$X(g,t,t')_j^{(N)} = \frac{1}{N} \sum_{i=1}^{N} X(\boldsymbol{\mathcal{T}}^{(i)}, G^{(i)}, g, t, t', \boldsymbol{\theta})_j, \tag{64}$$

$$Y(g,t,t')_j^{(N)} = \frac{1}{N} \sum_{i=1}^{N} Y(\boldsymbol{\mathcal{T}}^{(i)}, G^{(i)}, g, t, t', \boldsymbol{\theta})_j, \tag{65}$$

$$X(\boldsymbol{\tau}, g', g, t, t', \boldsymbol{\theta})_j = \left[ \prod_{k=1}^{t'-1} \frac{p(a_k \mid s_k, g, \boldsymbol{\theta})}{p(a_k \mid s_k, g', \boldsymbol{\theta})} \right] \frac{\partial}{\partial \theta_j} \log p(a_t \mid s_t, g, \boldsymbol{\theta}) r(s_{t'}, g), \tag{66}$$

$$Y(\boldsymbol{\tau}, g', g, t, t', \boldsymbol{\theta})_j = \left[ \prod_{k=1}^{t'-1} \frac{p(a_k \mid s_k, g, \boldsymbol{\theta})}{p(a_k \mid s_k, g', \boldsymbol{\theta})} \right]. \tag{67}$$

Consider the expected value $E_{X_i} = \mathbb{E}\left[ X(\boldsymbol{\mathcal{T}}^{(i)}, G^{(i)}, g, t, t', \boldsymbol{\theta})_j \mid \boldsymbol{\theta} \right]$, which is given by

$$E_{X_i} = \sum_{g^{(i)}} p(g^{(i)}) \mathbb{E}\left[ \left[ \prod_{k=1}^{t'-1} \frac{p(A_k \mid S_k, g, \boldsymbol{\theta})}{p(A_k \mid S_k, G = g^{(i)}, \boldsymbol{\theta})} \right] \frac{\partial}{\partial \theta_j} \log p(A_t \mid S_t, g, \boldsymbol{\theta}) r(S_{t'}, g) \mid G = g^{(i)}, \boldsymbol{\theta} \right]. \tag{68}$$

Using the fact that $t' > t$, Lemma D.1, and canceling terms, $E_{X_i}$ can be written as

$$\sum_{g^{(i)}} p(g^{(i)}) \sum_{s_{1:t'}} \sum_{a_{1:t'-1}} p(s_{t'} \mid s_{1:t'-1}, a_{1:t'-1}, G = g^{(i)}, \boldsymbol{\theta}) p(s_{1:t'-1}, a_{1:t'-1} \mid g, \boldsymbol{\theta}) \frac{\partial}{\partial \theta_j} \log p(a_t \mid s_t, g, \boldsymbol{\theta}) r(s_{t'}, g). \tag{69}$$

Because $S_{t'} \perp\!\!\!\perp G \mid S_{1:t'-1}, A_{1:t'-1}, \boldsymbol{\Theta}$,

$$E_{X_i} = \mathbb{E}\left[ \frac{\partial}{\partial \theta_j} \log p(A_t \mid S_t, g, \boldsymbol{\theta}) r(S_{t'}, g) \mid g, \boldsymbol{\theta} \right]. \tag{70}$$

Conditionally on $\boldsymbol{\Theta}$, the variable $X(g,t,t')_j^{(N)}$ is an average of iid random variables with expected value $E_{X_i}$. By the strong law of large numbers (Sen & Singer, 1994, Theorem 2.3.13), $X(g,t,t')_j^{(N)} \xrightarrow{\text{a.s.}} E_{X_i}$.

Using Lemma D.1, the expected value $E_{Y_i} = \mathbb{E}\left[ Y(\boldsymbol{\mathcal{T}}^{(i)}, G^{(i)}, g, t, t', \boldsymbol{\theta})_j \mid \boldsymbol{\theta} \right]$ is given by

$$E_{Y_i} = \sum_{g^{(i)}} p(g^{(i)}) \mathbb{E}\left[ \frac{p(S_{1:t'-1}^{(i)}, A_{1:t'-1}^{(i)} \mid G^{(i)} = g, \boldsymbol{\theta})}{p(S_{1:t'-1}^{(i)}, A_{1:t'-1}^{(i)} \mid g^{(i)}, \boldsymbol{\theta})} \mid g^{(i)}, \boldsymbol{\theta} \right] = 1. \tag{71}$$

Conditionally on $\boldsymbol{\Theta}$, the variable $Y(g,t,t')_j^{(N)}$ is an average of iid random variables with expected value 1. By the strong law of large numbers, $Y(g,t,t')_j^{(N)} \xrightarrow{\text{a.s.}} 1$.

Because both $X(g,t,t')_j^{(N)}$ and $Y(g,t,t')_j^{(N)}$ converge almost surely to real numbers (Thomas, 2015, Ch. 3, Property 2),

$$\frac{X(g,t,t')_j^{(N)}}{Y(g,t,t')_j^{(N)}} \xrightarrow{\text{a.s.}} \mathbb{E}\left[ \frac{\partial}{\partial \theta_j} \log p(A_t \mid S_t, g, \boldsymbol{\theta}) r(S_{t'}, g) \mid g, \boldsymbol{\theta} \right]. \tag{72}$$

By Theorem 3.1 and the fact that $W_j^{(N)}$ is a linear combination of terms $X(g, t, t')_j^{(N)}/Y(g, t, t')_j^{(N)}$,

$$W_j^{(N)} \xrightarrow{\text{a.s.}} \sum_g p(g) \sum_{t=1}^{T-1} \sum_{t'=t+1}^{T} \mathbb{E}\left[\frac{\partial}{\partial \theta_j} \log p(A_t \mid S_t, g, \boldsymbol{\theta}) r(S_{t'}, g) \mid g, \boldsymbol{\theta}\right] = \frac{\partial}{\partial \theta_j} \eta(\boldsymbol{\theta}). \quad (73)$$

$\square$

## C.3 THEOREM C.1

**Theorem C.1.** *The weighted baseline estimator, given by*

$$\sum_{i=1}^{N} \sum_g p(g) \sum_{t=1}^{T-1} \nabla \log p(A_t^{(i)} \mid S_t^{(i)}, G^{(i)} = g, \boldsymbol{\theta}) \frac{\left[\prod_{k=1}^{t} \frac{p(A_k^{(i)} \mid S_k^{(i)}, G^{(i)}=g, \boldsymbol{\theta})}{p(A_k^{(i)} \mid S_k^{(i)}, G^{(i)}, \boldsymbol{\theta})}\right] b_t^{\boldsymbol{\theta}}(S_t^{(i)}, g)}{\sum_{j=1}^{N}\left[\prod_{k=1}^{t} \frac{p(A_k^{(j)} \mid S_k^{(j)}, G^{(j)}=g, \boldsymbol{\theta})}{p(A_k^{(j)} \mid S_k^{(j)}, G^{(j)}, \boldsymbol{\theta})}\right]}, \quad (74)$$

*converges almost surely to zero.*

*Proof.* Let $B_j^{(N)}$ denote the $j$-th element of the estimator, which can be written as

$$B_j^{(N)} = \sum_g p(g) \sum_{t=1}^{T-1} \frac{X(g, t)_j^{(N)}}{Y(g, t)_j^{(N)}}, \quad (75)$$

where

$$X(g, t)_j^{(N)} = \frac{1}{N} \sum_{i=1}^{N} X(\boldsymbol{\mathcal{T}}^{(i)}, G^{(i)}, g, t, \boldsymbol{\theta})_j, \quad (76)$$

$$Y(g, t)_j^{(N)} = \frac{1}{N} \sum_{i=1}^{N} Y(\boldsymbol{\mathcal{T}}^{(i)}, G^{(i)}, g, t, \boldsymbol{\theta})_j, \quad (77)$$

$$X(\boldsymbol{\tau}, g', g, t, \boldsymbol{\theta})_j = \left[\prod_{k=1}^{t} \frac{p(a_k \mid s_k, g, \boldsymbol{\theta})}{p(a_k \mid s_k, g', \boldsymbol{\theta})}\right] \frac{\partial}{\partial \theta_j} \log p(a_t \mid s_t, g, \boldsymbol{\theta}) b_t^{\boldsymbol{\theta}}(s_t, g), \quad (78)$$

$$Y(\boldsymbol{\tau}, g', g, t, \boldsymbol{\theta})_j = \prod_{k=1}^{t} \frac{p(a_k \mid s_k, g, \boldsymbol{\theta})}{p(a_k \mid s_k, g', \boldsymbol{\theta})}. \quad (79)$$

Using Eqs. 44 and 47, the expected value $E_{X_i} = \mathbb{E}\left[X(\boldsymbol{\mathcal{T}}^{(i)}, G^{(i)}, g, t, \boldsymbol{\theta})_j \mid \boldsymbol{\theta}\right]$ is given by

$$E_{X_i} = \sum_{g^{(i)}} p(g^{(i)}) \mathbb{E}\left[X(\boldsymbol{\mathcal{T}}^{(i)}, g^{(i)}, g, t, \boldsymbol{\theta})_j \mid g^{(i)}, \boldsymbol{\theta}\right] = 0. \quad (80)$$

Conditionally on $\boldsymbol{\Theta}$, the variable $X(g, t)_j^{(N)}$ is an average of iid random variables with expected value zero. By the strong law of large numbers (Sen & Singer, 1994, Theorem 2.3.13), $X(g, t)_j^{(N)} \xrightarrow{\text{a.s.}} 0$.

The fact that $Y(g, t)_j^{(N)} \xrightarrow{\text{a.s.}} 1$ is already established in the proof of Theorem 5.2. Because both $X(g, t)_j^{(N)}$ and $Y(g, t)_j^{(N)}$ converge almost surely to real numbers (Thomas, 2015, Ch. 3, Property 2),

$$\frac{X(g, t)_j^{(N)}}{Y(g, t)_j^{(N)}} \xrightarrow{\text{a.s.}} 0. \quad (81)$$

Because $B_j^{(N)}$ is a linear combination of terms $X(g, t)_j^{(N)}/Y(g, t)_j^{(N)}$, $B_j^{(N)} \xrightarrow{\text{a.s.}} 0$.

$\square$

Clearly, if $E^{(N)}$ is a consistent estimator of a some quantity given $\boldsymbol{\theta}$, then so is $E^{(N)} - B_j^{(N)}$, which allows using this result in combination with Theorem 5.2.

# D FUNDAMENTAL RESULTS

This appendix presents results required by previous sections: Lemma D.1 (App. D.1), Lemma D.2 (App. D.2), Theorem 4.4 (App. D.4), and Lemma D.4 (App. D.5). Appendix D.3 contains an auxiliary result.

## D.1 LEMMA D.1

**Lemma D.1.** *For every $\tau, g, \boldsymbol{\theta}$, and $1 \leq t \leq T - 1$,*

$$p(s_{1:t}, a_{1:t} \mid g, \boldsymbol{\theta}) = p(s_1)p(a_t \mid s_t, g, \boldsymbol{\theta}) \prod_{k=1}^{t-1} p(a_k \mid s_k, g, \boldsymbol{\theta})p(s_{k+1} \mid s_k, a_k). \tag{82}$$

*Proof.* In order to employ backward induction, consider the case $t = T - 1$. By marginalization,

$$p(s_{1:T-1}, a_{1:T-1} \mid g, \boldsymbol{\theta}) = \sum_{s_T} p(\tau \mid g, \boldsymbol{\theta}) = \sum_{s_T} p(s_1) \prod_{k=1}^{T-1} p(a_k \mid s_k, g, \boldsymbol{\theta})p(s_{k+1} \mid s_k, a_k) \tag{83}$$

$$= p(s_1)p(a_{T-1} \mid s_{T-1}, g, \boldsymbol{\theta}) \prod_{k=1}^{T-2} p(a_k \mid s_k, g, \boldsymbol{\theta})p(s_{k+1} \mid s_k, a_k), \tag{84}$$

which completes the proof of the base case.

Assuming the inductive hypothesis is true for a given $2 \leq t \leq T - 1$ and considering the case $t - 1$,

$$p(s_{1:t-1}, a_{1:t-1} \mid g, \boldsymbol{\theta}) = \sum_{s_t} \sum_{a_t} p(s_1)p(a_t \mid s_t, g, \boldsymbol{\theta}) \prod_{k=1}^{t-1} p(a_k \mid s_k, g, \boldsymbol{\theta})p(s_{k+1} \mid s_k, a_k) \tag{85}$$

$$= p(s_1)p(a_{t-1} \mid s_{t-1}, g, \boldsymbol{\theta}) \prod_{k=1}^{t-2} p(a_k \mid s_k, g, \boldsymbol{\theta})p(s_{k+1} \mid s_k, a_k). \tag{86}$$

$\square$

## D.2 LEMMA D.2

**Lemma D.2.** *For every $\tau, g, \boldsymbol{\theta}$, and $1 \leq t \leq T$,*

$$p(s_{t:T}, a_{t:T-1} \mid s_{1:t-1}, a_{1:t-1}, g, \boldsymbol{\theta}) = p(s_t \mid s_{t-1}, a_{t-1}) \prod_{k=t}^{T-1} p(a_k \mid s_k, g, \boldsymbol{\theta})p(s_{k+1} \mid s_k, a_k). \tag{87}$$

*Proof.* The case $t = 1$ can be inspected easily. Consider $2 \leq t \leq T$. By definition,

$$p(s_{t:T}, a_{t:T-1} \mid s_{1:t-1}, a_{1:t-1}, g, \boldsymbol{\theta}) = \frac{p(s_{1:T}, a_{1:T-1} \mid g, \boldsymbol{\theta})}{p(s_{1:t-1}, a_{1:t-1} \mid g, \boldsymbol{\theta})}. \tag{88}$$

Using Lemma D.1,

$$p(s_{t:T}, a_{t:T-1} \mid s_{1:t-1}, a_{1:t-1}, g, \boldsymbol{\theta}) = \frac{p(s_1) \prod_{k=1}^{T-1} p(a_k \mid s_k, g, \boldsymbol{\theta})p(s_{k+1} \mid s_k, a_k)}{p(s_1)p(a_{t-1} \mid s_{t-1}, g, \boldsymbol{\theta}) \prod_{k=1}^{t-2} p(a_k \mid s_k, g, \boldsymbol{\theta})p(s_{k+1} \mid s_k, a_k)} \tag{89}$$

$$= \frac{\prod_{k=t-1}^{T-1} p(a_k \mid s_k, g, \boldsymbol{\theta})p(s_{k+1} \mid s_k, a_k)}{p(a_{t-1} \mid s_{t-1}, g, \boldsymbol{\theta})}. \tag{90}$$

$\square$

### D.3 LEMMA D.3

**Lemma D.3.** *For every $t$ and $\boldsymbol{\theta}$, the action-value function $Q_t^{\boldsymbol{\theta}}$ is given by*

$$Q_t^{\boldsymbol{\theta}}(s, a, g) = \mathbb{E}\left[ r(S_{t+1}, g) + V_{t+1}^{\boldsymbol{\theta}}(S_{t+1}, g) \mid S_t = s, A_t = a \right]. \tag{91}$$

*Proof.* From the definition of action-value function and using the fact that $S_{t+1} \perp\!\!\!\perp G, \boldsymbol{\Theta} \mid S_t, A_t$,

$$Q_t^{\boldsymbol{\theta}}(s, a, g) = \mathbb{E}\left[ r(S_{t+1}, g) \mid S_t = s, A_t = a \right] + \mathbb{E}\left[ \sum_{t'=t+2}^{T} r(S_{t'}, g) \mid S_t = s, A_t = a, g, \boldsymbol{\theta} \right]. \tag{92}$$

Let $z$ denote the second term in the right-hand side of the previous equation, which can also be written as

$$z = \sum_{s_{t+1}} \sum_{a_{t+1}} \sum_{s_{t+2:T}} p(s_{t+1}, a_{t+1}, s_{t+2:T} \mid S_t = s, A_t = a, g, \boldsymbol{\theta}) \sum_{t'=t+2}^{T} r(s_{t'}, g). \tag{93}$$

Consider the following three independence properties:

$$S_{t+1} \perp\!\!\!\perp G, \boldsymbol{\Theta} \mid S_t, A_t, \tag{94}$$
$$A_{t+1} \perp\!\!\!\perp S_t, A_t \mid S_{t+1}, G, \boldsymbol{\Theta}, \tag{95}$$
$$S_{t+2:T} \perp\!\!\!\perp S_t, A_t \mid S_{t+1}, A_{t+1}, G, \boldsymbol{\Theta}. \tag{96}$$

Together, these properties can be used to demonstrate that

$$z = \sum_{s_{t+1}} p(s_{t+1} \mid S_t = s, A_t = a) \sum_{a_{t+1}} p(a_{t+1} \mid s_{t+1}, g, \boldsymbol{\theta}) \sum_{s_{t+2:T}} p(s_{t+2:T} \mid s_{t+1}, a_{t+1}, g, \boldsymbol{\theta}) \sum_{t'=t+2}^{T} r(s_{t'}, g). \tag{97}$$

From the definition of value function, $z = \mathbb{E}\left[ V_{t+1}^{\boldsymbol{\theta}}(S_{t+1}, g) \mid S_t = s, A_t = a \right]$.

$\square$

### D.4 THEOREM 4.4

**Theorem 4.4.** *For every $t$ and $\boldsymbol{\theta}$, the advantage function $A_t^{\boldsymbol{\theta}}$ is given by*

$$A_t^{\boldsymbol{\theta}}(s, a, g) = \mathbb{E}\left[ r(S_{t+1}, g) + V_{t+1}^{\boldsymbol{\theta}}(S_{t+1}, g) - V_t^{\boldsymbol{\theta}}(s, g) \mid S_t = s, A_t = a \right]. \tag{10}$$

*Proof.* The result follows from the definition of advantage function and Lemma D.3. $\square$

### D.5 LEMMA D.4

Consider a discrete random variable $X$ and real-valued functions $f$ and $h$. Suppose also that $\mathbb{E}\left[ h(X) \right] = 0$ and $\mathrm{Var}\left[ h(X) \right] > 0$. Clearly, for every $b \in \mathbb{R}$, we have $\mathbb{E}\left[ f(X) - bh(X) \right] = \mathbb{E}\left[ f(X) \right]$.

**Lemma D.4.** *The constant $b \in \mathbb{R}$ that minimizes $\mathrm{Var}\left[ f(X) - bh(X) \right]$ is given by*

$$b = \frac{\mathbb{E}\left[ f(X)h(X) \right]}{\mathbb{E}\left[ h(X)^2 \right]}. \tag{98}$$

*Proof.* Let $v = \mathrm{Var}\left[ f(X) - bh(X) \right]$. Using our assumptions and the definition of variance,

$$v = \mathbb{E}\left[ (f(X) - bh(X))^2 \right] - \mathbb{E}\left[ f(X) - bh(X) \right]^2 = \mathbb{E}\left[ (f(X) - bh(X))^2 \right] - \mathbb{E}\left[ f(X) \right]^2 \tag{99}$$
$$= \mathbb{E}\left[ f(X)^2 \right] - 2b\mathbb{E}\left[ f(X)h(X) \right] + b^2\mathbb{E}\left[ h(X)^2 \right] - \mathbb{E}\left[ f(X) \right]^2. \tag{100}$$

The first and second derivatives of $v$ with respect to $b$ are given by $dv/db = -2\mathbb{E}\left[f(X)h(X)\right] + 2b\mathbb{E}\left[h(X)^2\right]$ and $d^2v/db^2 = 2\mathbb{E}\left[h(X)^2\right]$. Our assumptions guarantee that $\mathbb{E}\left[h(X)^2\right] > 0$. Therefore, by Fermat's theorem, if $b$ is a local minimum, then $dv/db = 0$, leading to the desired equality. By the second derivative test, $b$ must be a local minimum.

$\square$

# E EXPERIMENTS

This appendix contains additional information about the experiments introduced in Section 6. Appendix E.1 details policy and baseline representations. Appendix E.2 documents experimental settings. Appendix E.3 presents unabridged results.

## E.1 POLICY AND BASELINE REPRESENTATIONS

In every experiment, a policy is represented by a feedforward neural network with a *softmax* output layer. The input to such a policy is a pair composed of state and goal. A baseline function is represented by a feedforward neural network with a single (linear) output neuron. The input to such a baseline function is a triple composed of state, goal, and time step. The baseline function is trained to approximate the value function using the mean squared (one-step) temporal difference error (Sutton & Barto, 1998). Parameters are updated using Adam (Kingma & Ba, 2014). The networks are given by the following.

**Bit flipping environments and grid world environments.** Both policy and baseline networks have two hidden layers, each with 256 hyperbolic tangent units. Every weight is initially drawn from a Gaussian distribution with mean 0 and standard deviation 0.01 (and redrawn if far from the mean by two standard deviations), and every bias is initially zero.

**Ms. Pac-man environment.** The policy network is represented by a convolutional neural network. The network architecture is given by a convolutional layer with 32 filters ($8\times8$, stride 4); convolutional layer with 64 filters ($4 \times 4$, stride 2); convolutional layer with 64 filters ($3 \times 3$, stride 1); and three fully-connected layers, each with 256 units. Every unit uses a hyperbolic tangent activation function. Every weight is initially set using variance scaling (Glorot & Bengio, 2010), and every bias is initially zero. These design decisions are similar to the ones made by Mnih et al. (2015).

A sequence of images obtained from the Arcade Learning Environment (Bellemare et al., 2013) is preprocessed as follows. Individually for each color channel, an elementwise maximum operation is employed between two consecutive images to reduce rendering artifacts. Such $210 \times 160 \times 3$ preprocessed image is converted to grayscale, cropped, and rescaled into an $84 \times 84$ image $x_t$. A sequence of images $x_{t-12}, x_{t-8}, x_{t-4}, x_t$ obtained in this way is *stacked* into an $84 \times 84 \times 4$ image, which is an input to the policy network (recall that each action is repeated for 13 game ticks). The goal information is concatenated with the *flattened* output of the last convolutional layer.

**FetchPush environment.** The policy network has three hidden layers, each with 256 hyperbolic tangent units. Every weight is initially set using variance scaling (Glorot & Bengio, 2010), and every bias is initially zero.

## E.2 EXPERIMENTAL SETTINGS

Tables 1 and 2 document the experimental settings. The number of runs, training batches, and batches between evaluations are reported separately for hyperparameter search and definitive runs. The number of training batches is adapted according to how soon each estimator leads to apparent convergence. Note that it is very difficult to establish this setting before hyperparameter search. The number of batches between evaluations is adapted so that there are 100 evaluation steps in total.

Other settings include the sets of policy and baseline learning rates under consideration for hyperparameter search, and the number of active goals subsampled per episode. In Tables 1 and 2, $\mathcal{R}_1 = \{\alpha \times 10^{-k} \mid \alpha \in \{1, 5\} \text{ and } k \in \{2, 3, 4, 5\}\}$ and $\mathcal{R}_2 = \{\beta \times 10^{-5} \mid \beta \in \{1, 2.5, 5, 7.5, 10\}\}$.

As already mentioned in Section 6, the definitive runs use the best combination of hyperparameters (learning rates) found for each estimator. Every setting was carefully chosen during preliminary experiments to ensure that the best result for each estimator is representative. In particular, the best performing learning rates rarely lie on the extrema of the corresponding search range. In the single case where the best performing learning rate found by hyperparameter search for a goal-conditional policy gradient estimator was such an extreme value (FetchPush, for a small batch size), evaluating one additional learning rate lead to decreased average performance.

Table 1: Experimental settings for the bit flipping and grid world environments

| | Bit flipping (8 bits) | | Bit flipping (16 bits) | |
| --- | --- | --- | --- | --- |
| | Batch size 2 | Batch size 16 | Batch size 2 | Batch size 16 |
| Runs (definitive) | 20 | 20 | 20 | 20 |
| Training batches (definitive) | 5000 | 1400 | 15000 | 1000 |
| Batches between evaluations (definitive) | 50 | 14 | 150 | 10 |
| Runs (search) | 10 | 10 | 10 | 10 |
| Training batches (search) | 4000 | 1400 | 4000 | 1000 |
| Batches between evaluations (search) | 40 | 14 | 40 | 10 |
| Policy learning rates | $\mathcal{R}_1$ | $\mathcal{R}_1$ | $\mathcal{R}_1$ | $\mathcal{R}_1$ |
| Baseline learning rates | $\mathcal{R}_1$ | $\mathcal{R}_1$ | $\mathcal{R}_1$ | $\mathcal{R}_1$ |
| Episodes per evaluation | 256 | 256 | 256 | 256 |
| Maximum active goals per episode | $\infty$ | $\infty$ | $\infty$ | $\infty$ |

| | Empty room | | Four rooms | |
| --- | --- | --- | --- | --- |
| | Batch size 2 | Batch size 16 | Batch size 2 | Batch size 16 |
| Runs (definitive) | 20 | 20 | 20 | 20 |
| Training batches (definitive) | 2200 | 200 | 10000 | 1700 |
| Batches between evaluations (definitive) | 22 | 2 | 100 | 17 |
| Runs (search) | 10 | 10 | 10 | 10 |
| Training batches (search) | 2500 | 800 | 10000 | 3500 |
| Batches between evaluations (search) | 25 | 8 | 100 | 35 |
| Policy learning rates | $\mathcal{R}_1$ | $\mathcal{R}_1$ | $\mathcal{R}_1$ | $\mathcal{R}_1$ |
| Baseline learning rates | $\mathcal{R}_1$ | $\mathcal{R}_1$ | $\mathcal{R}_1$ | $\mathcal{R}_1$ |
| Episodes per evaluation | 256 | 256 | 256 | 256 |
| Maximum active goals per episode | $\infty$ | $\infty$ | $\infty$ | $\infty$ |

Table 2: Experimental settings for the Ms. Pac-man and FetchPush environments

| | Ms. Pac-man | | FetchPush | |
|---|---|---|---|---|
| | Batch size 2 | Batch size 16 | Batch size 2 | Batch size 16 |
| Runs (definitive) | 10 | 10 | 10 | 10 |
| Training batches (definitive) | 40000 | 12500 | 40000 | 12500 |
| Batches between evaluations (definitive) | 400 | 125 | 400 | 125 |
| Runs (search) | 5 | 5 | 5 | 5 |
| Training batches (search) | 40000 | 12000 | 40000 | 15000 |
| Batches between evaluations (search) | 800 | 120 | 800 | 300 |
| *Policy learning rates* | $\mathcal{R}_2$ | $\mathcal{R}_2$ | $\mathcal{R}_2$ | $\mathcal{R}_2$ |
| Episodes per evaluation | 240 | 240 | 512 | 512 |
| Maximum active goals per episode | $\infty$ | 3 | $\infty$ | 3 |

### E.3 RESULTS

This appendix contains unabridged experimental results. Appendices E.3.1 and E.3.2 present hyperparameter sensitivity plots for every combination of environment and batch size. A hyperparameter sensitivity plot displays the average performance achieved by each hyperparameter setting (sorted from best to worst along the horizontal axis). Appendices E.3.3 and E.3.4 present learning curves for every combination of environment and batch size. Appendix E.3.5 presents average performance results. Appendix E.3.6 presents an empirical study of likelihood ratios. Appendix E.3.7 presents an empirical comparison with hindsight experience replay (Andrychowicz et al., 2017).

#### E.3.1 HYPERPARAMETER SENSITIVITY PLOTS (BATCH SIZE 2)

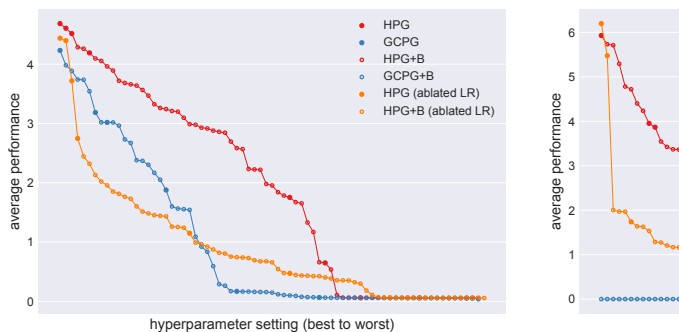

Figure 10: Bit flipping ($k = 8$).

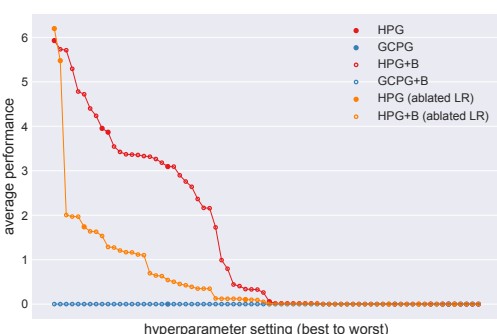

Figure 11: Bit flipping ($k = 16$).

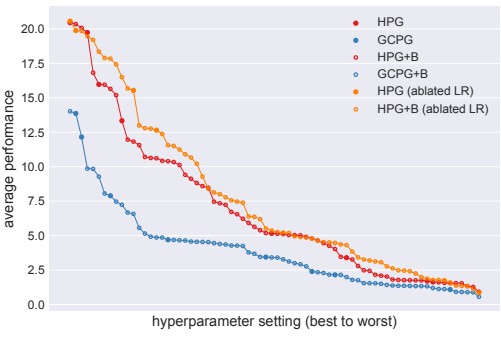

Figure 12: Empty room.

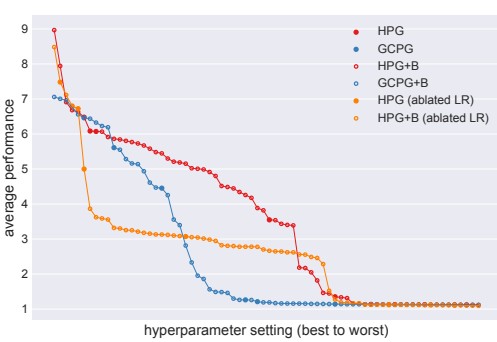

Figure 13: Four rooms.

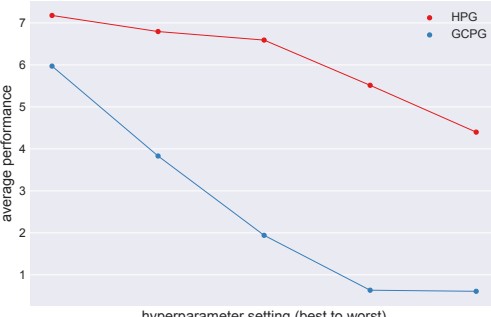

Figure 14: Ms. Pac-man.

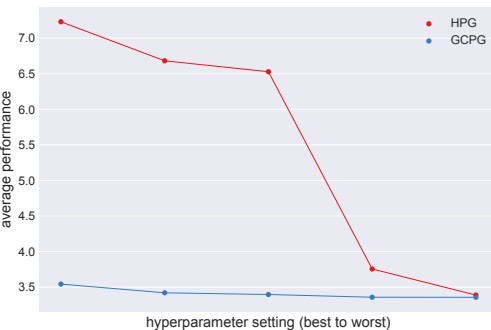

Figure 15: FetchPush.

### E.3.2 HYPERPARAMETER SENSITIVITY PLOTS (BATCH SIZE 16)

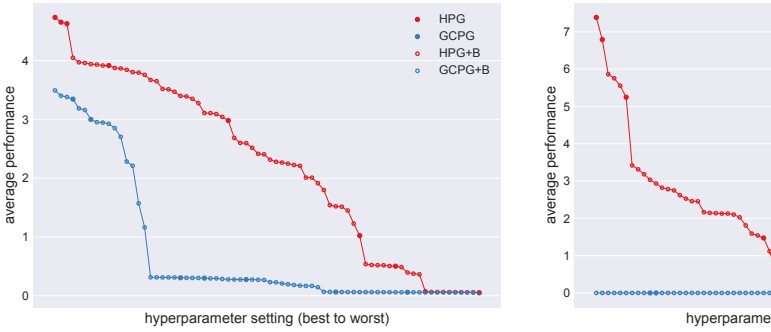

Figure 16: Bit flipping ($k = 8$).

Figure 17: Bit flipping ($k = 16$).

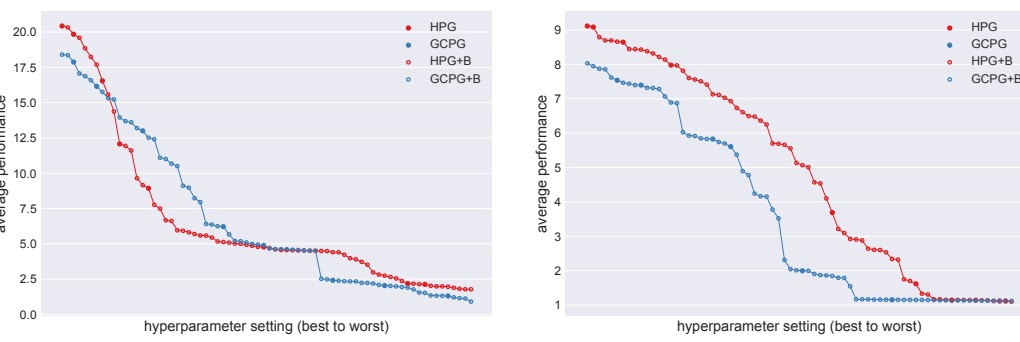

Figure 18: Empty room.

Figure 19: Four rooms.

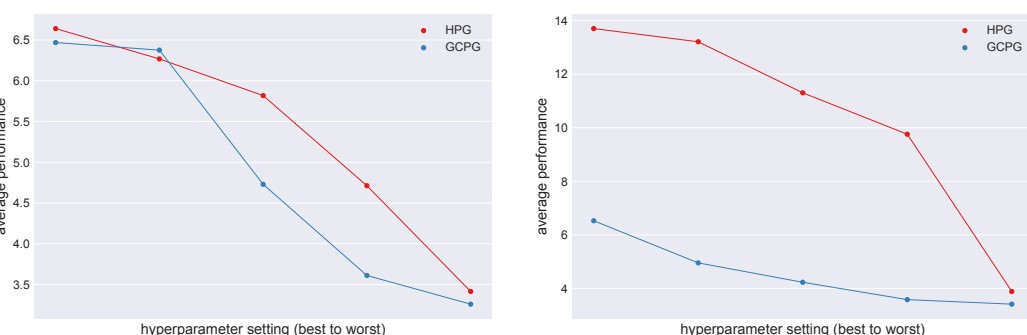

Figure 20: Ms. Pac-man.

Figure 21: FetchPush.

### E.3.3 LEARNING CURVES (BATCH SIZE 2)

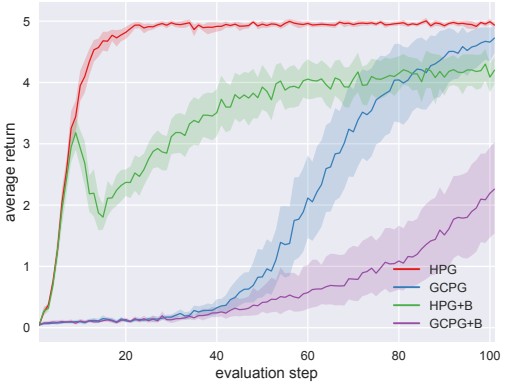

Figure 22: Bit flipping ($k = 8$).

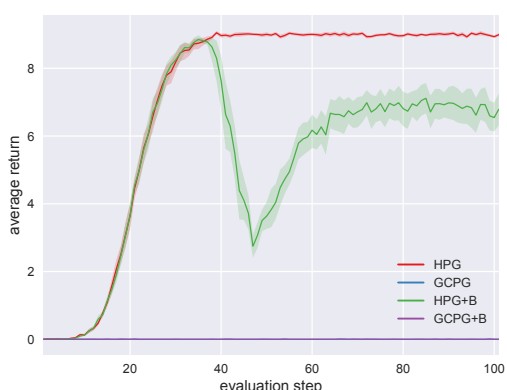

Figure 23: Bit flipping ($k = 16$).

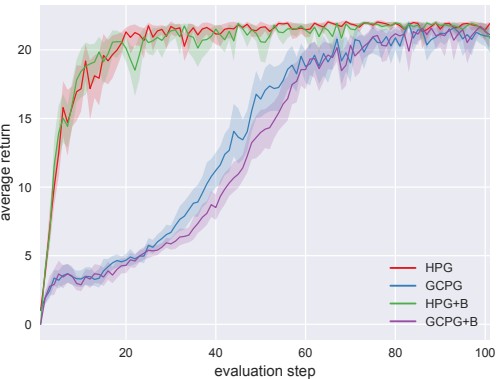

Figure 24: Empty room.

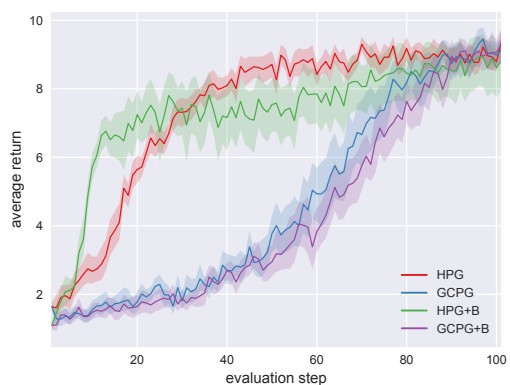

Figure 25: Four rooms.

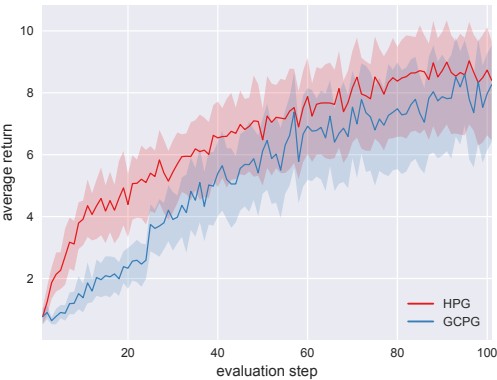

Figure 26: Ms. Pac-man.

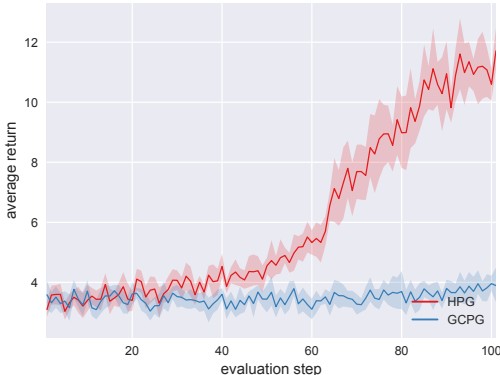

Figure 27: FetchPush.

### E.3.4 Learning curves (batch size 16)

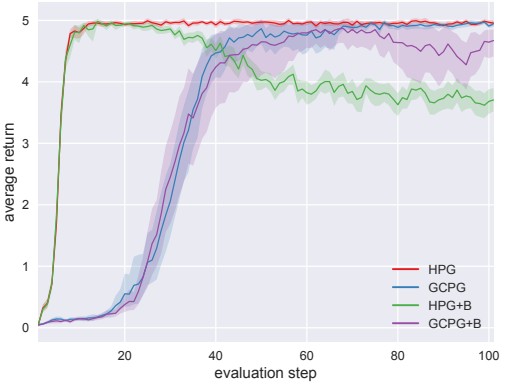

Figure 28: Bit flipping ($k = 8$).

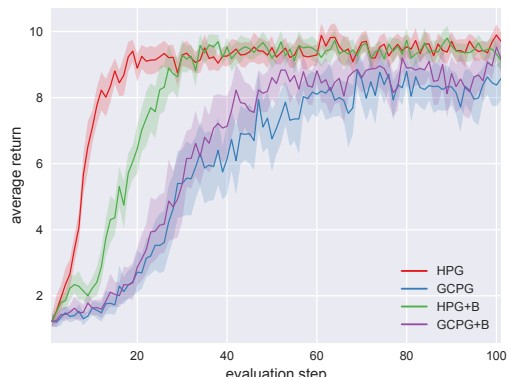

Figure 29: Bit flipping ($k = 16$).

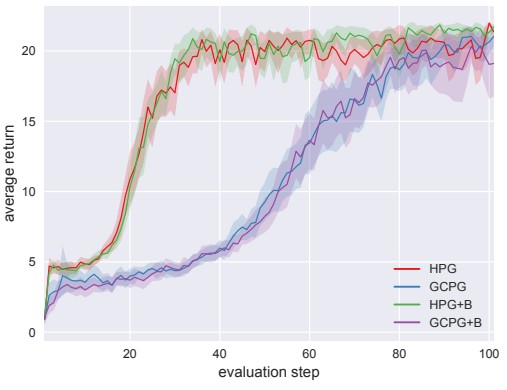

Figure 30: Empty room.

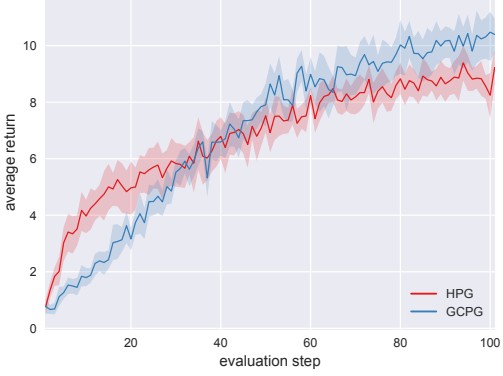

Figure 31: Four rooms.

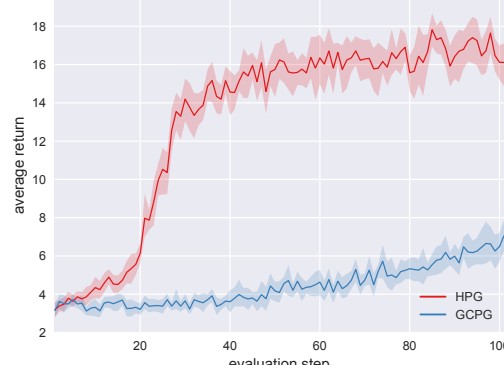

Figure 32: Ms. Pac-man.

Figure 33: FetchPush.

### E.3.5 Average performance results

Table 3 presents average performance results for every combination of environment and batch size.

Table 3: Definitive average performance results

|  | Bit flipping (8 bits) | | Bit flipping (16 bits) | |
| --- | --- | --- | --- | --- |
|  | Batch size 2 | Batch size 16 | Batch size 2 | Batch size 16 |
| HPG | $\mathbf{4.60 \pm 0.06}$ | $\mathbf{4.72 \pm 0.02}$ | $\mathbf{7.11 \pm 0.12}$ | $\mathbf{7.39 \pm 0.24}$ |
| GCPG | $1.81 \pm 0.61$ | $3.44 \pm 0.30$ | $0.00 \pm 0.00$ | $0.00 \pm 0.00$ |
| HPG+B | $3.40 \pm 0.46$ | $4.04 \pm 0.10$ | $5.35 \pm 0.40$ | $6.09 \pm 0.29$ |
| GCPG+B | $0.64 \pm 0.58$ | $3.31 \pm 0.58$ | $0.00 \pm 0.00$ | $0.00 \pm 0.00$ |

|  | Empty room | | Four rooms | |
| --- | --- | --- | --- | --- |
|  | Batch size 2 | Batch size 16 | Batch size 2 | Batch size 16 |
| HPG | $\mathbf{20.22 \pm 0.37}$ | $16.83 \pm 0.84$ | $\mathbf{7.38 \pm 0.16}$ | $\mathbf{8.75 \pm 0.12}$ |
| GCPG | $12.54 \pm 1.01$ | $10.96 \pm 1.24$ | $4.64 \pm 0.57$ | $6.12 \pm 0.54$ |
| HPG+B | $19.90 \pm 0.29$ | $\mathbf{17.12 \pm 0.44}$ | $7.28 \pm 1.28$ | $8.08 \pm 0.18$ |
| GCPG+B | $12.69 \pm 1.16$ | $10.68 \pm 1.36$ | $4.26 \pm 0.55$ | $6.61 \pm 0.49$ |

|  | Ms. Pac-man | | FetchPush | |
| --- | --- | --- | --- | --- |
|  | Batch size 2 | Batch size 16 | Batch size 2 | Batch size 16 |
| HPG | $\mathbf{6.58 \pm 1.96}$ | $6.80 \pm 0.64$ | $\mathbf{6.10 \pm 0.34}$ | $\mathbf{13.15 \pm 0.40}$ |
| GCPG | $5.29 \pm 1.67$ | $\mathbf{6.92 \pm 0.58}$ | $3.48 \pm 0.15$ | $4.42 \pm 0.28$ |

### E.3.6 LIKELIHOOD RATIO PLOTS

This appendix presents a study of the *active* (normalized) likelihood ratios computed by agents during training. A likelihood ratio is considered active if and only if it multiplies a non-zero reward (see Expression 12). Note that only these likelihood ratios affect gradient estimates based on HPG.

This study is conveyed through plots that encode the distribution of active likelihood ratios computed during training, individually for each time step within an episode. Each plot corresponds to an agent that employs HPG and obtains the highest definitive average performance for a given environment (Figs. 34-39). Note that the length of the largest bar for a given time step is fixed to aid visualization.

The most important insight provided by these plots is that likelihood ratios behave very differently across environments, even for equivalent time steps (for instance, compare bit flipping environments to grid world environments). In contrast, after the first time step, the behavior of likelihood ratios changes slowly across time steps within the same environment. In any case, alternative goals have a significant effect on gradient estimates, which agrees with the results presented in Section 6.

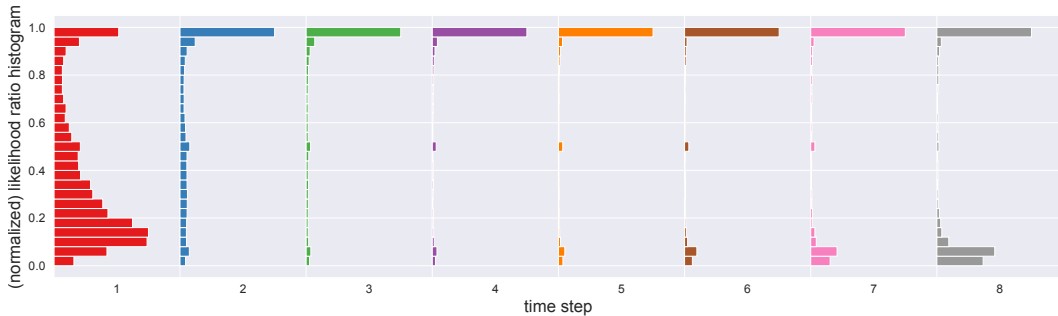

Figure 34: Bit flipping ($k = 8$, batch size 16).

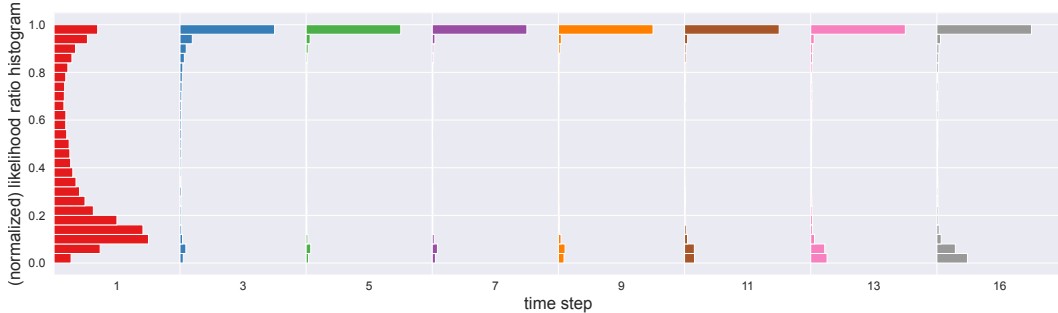

Figure 35: Bit flipping ($k = 16$, batch size 16).

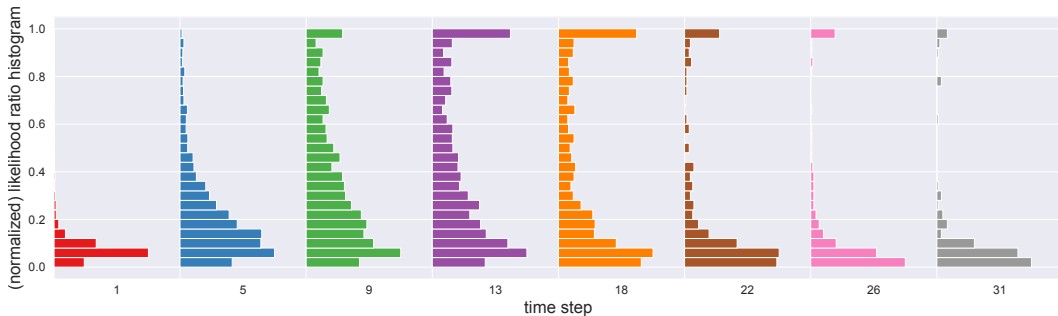

Figure 36: Empty room (batch size 16).

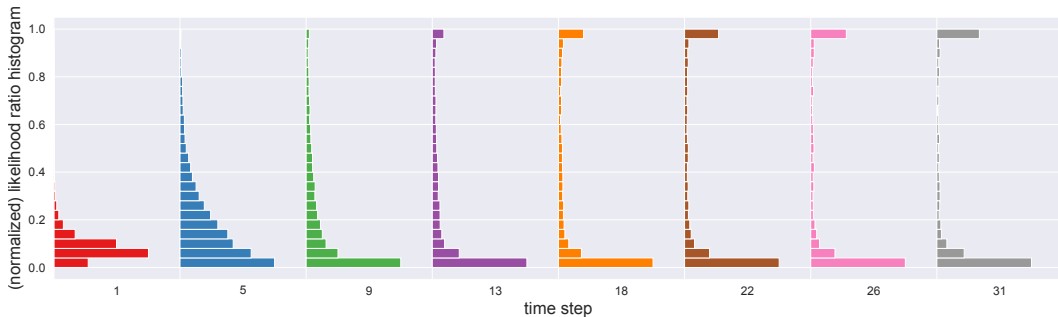

Figure 37: Four rooms (batch size 16).

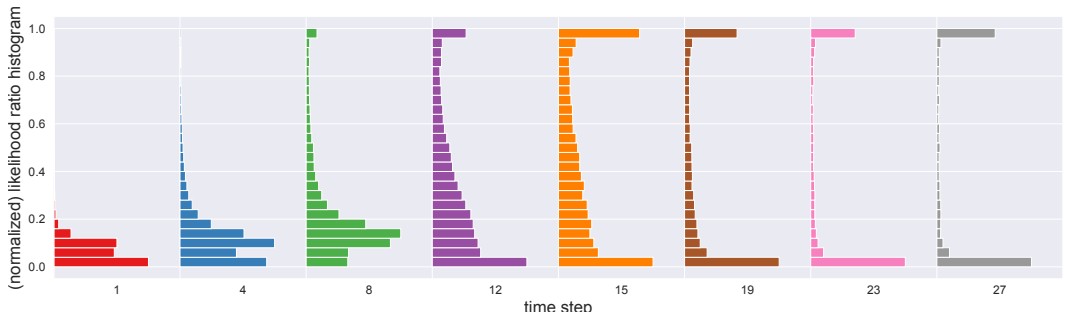

Figure 38: Ms. Pac-man (batch size 16).

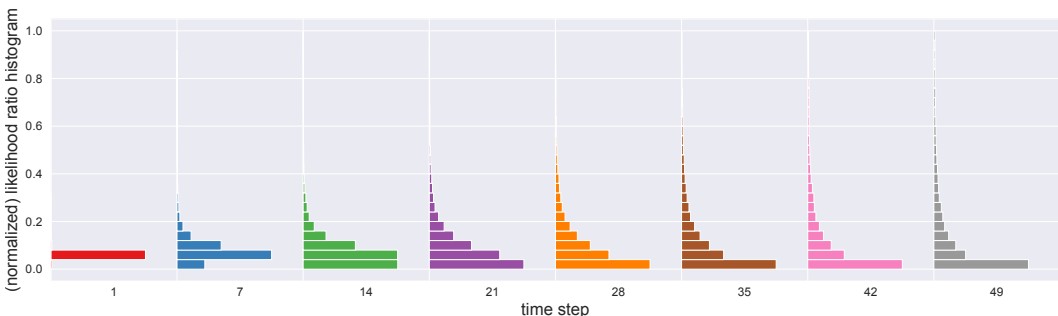

Figure 39: FetchPush (batch size 16).

### E.3.7 HINDSIGHT EXPERIENCE REPLAY

This appendix documents an empirical comparison between goal-conditional policy gradients (GCPG), hindsight policy gradients (HPG), deep Q-networks (Mnih et al., 2015, DQN), and a combination of DQN and hindsight experience replay (Andrychowicz et al., 2017, DQN+HER).

**Experience replay.** Our implementations of both DQN and DQN+HER are based on OpenAI Baselines (Dhariwal et al., 2017), and use mostly the same hyperparameters that Andrychowicz et al. (2017) used in their experiments on environments with discrete action spaces, all of which resemble our bit flipping environments. The only notable differences in our implementations are the lack of both Polyak-averaging and temporal difference target clipping.

Concretely, a *cycle* begins when an agent collects a number of episodes (16) by following an $\epsilon$-greedy policy derived from its deep Q-network ($\epsilon = 0.2$). The corresponding transitions are included in a replay buffer, which contains at most $10^6$ transitions. In the case of DQN+HER, hindsight transitions derived from a *final* strategy are also included in this replay buffer. Completing the cycle, for a total of 40 different batches, a batch composed of 128 transitions chosen at random from the replay buffer is used to define a loss function and allow one step of gradient-based minimization. The targets required to define these loss functions are computed using a copy of the deep Q-network from the start of the corresponding cycle. Parameters are updated using Adam (Kingma & Ba, 2014). A discount factor of $\gamma = 0.98$ is used, and seems necessary to improve the stability of both DQN and DQN+HER.

**Network architectures.** In every experiment, the deep Q-network is implemented by a feedforward neural network with a linear output neuron corresponding to each action. The input to such a network is a triple composed of state, goal, and time step. The network architectures are the same as those described in Appendix E.1, except that every weight is initially set using variance scaling (Glorot & Bengio, 2010), and all hidden layers use rectified linear units (Nair & Hinton, 2010). For the Ms. Pac-man environment, the time step information is concatenated with the flattened output of the last convolutional layer (together with the goal information). In comparison to the architecture employed by Andrychowicz et al. (2017) for environments with discrete action spaces, our architectures have one or two additional hidden layers (besides the convolutional architecture used for Ms. Pac-man).

**Experimental protocol.** The experimental protocol employed in our comparison is very similar to the one described in Section 6. Each agent is evaluated periodically, after a number of cycles that depends on the environment. During this evaluation, the agent collects a number of episodes by following a greedy policy derived from its deep Q-network.

For each environment, grid search is used to select the learning rates for both DQN and DQN+HER according to average performance scores (after the corresponding standard deviation across runs is subtracted, as in Section 6). The candidate sets of learning rates are the following. Bit flipping and grid world environments: $\{\alpha \times 10^{-k} \mid \alpha \in \{1, 5\} \text{ and } k \in \{2, 3, 4, 5\}\}$, FetchPush: $\{10^{-2}, 5 \times 10^{-3}, 10^{-3}, 5 \times 10^{-4}, 10^{-4}\}$, Ms. Pac-man: $\{10^{-3}, 5 \times 10^{-4}, 10^{-4}, 5 \times 10^{-5}, 10^{-5}\}$. These sets were carefully chosen such that the best performing learning rates do not lie on their extrema.

Definitive results for a given environment are obtained by using the best hyperparameters found for each method in additional runs. These definitive results are directly comparable to our previous results for GCPG and HPG (batch size 16), since every method will have interacted with the environment for the same number of episodes before each evaluation step. For each environment, the number of runs, the number of training batches (cycles), the number of batches (cycles) between evaluations, and the number of episodes per evaluation step are the same as those listed in Tables 1 and 2.

**Results.** The definitive results for the different environments are represented by learning curves (Figs. 40-45, Pg. 38). In the bit flipping environment for $k = 8$ (Figure 40), HPG and DQN+HER lead to equivalent sample efficiency, while GCPG lags far behind and DQN is completely unable to learn. In the bit flipping environment for $k = 16$ (Figure 41), HPG surpasses DQN+HER in sample efficiency by a small margin, while both GCPG and DQN are completely unable to learn. In the empty room environment (Figure 42), HPG is arguably the most sample efficient method, although DQN+HER is more stable upon obtaining a good policy. GCPG eventually obtains a good policy, whereas DQN exhibits instability. In the four rooms environment (Figure 43), DQN+HER outperforms all other methods by a large margin. Although DQN takes much longer to obtain good

policies, it would likely surpass both HPG and GCPG given additional training cycles. In the Ms. Pac-man environment (Figure 44), DQN+HER once again outperforms all other methods, which achieve equivalent sample efficiency (although DQN appears unstable by the end of training). In the FetchPush environment (Figure 45), HPG dramatically outperforms all other methods. Both DQN+HER and DQN are completely unable to learn, while GCPG appears to start learning by the end of the training process. Note that active goals are harshly subsampled to increase the computational efficiency of HPG for both Ms. Pac-man and FetchPush (see Sec. 6.3 and Sec. 6.4).

**Discussion.** Our results suggest that the decision between applying HPG or DQN+HER in a particular sparse-reward environment requires experimentation. In contrast, the decision to apply hindsight was always successful.

Note that we have not employed heuristics that are known to sometimes increase the performance of policy gradient methods (such as entropy bonuses, reward scaling, learning rate annealing, and simple statistical baselines) to avoid introducing confounding factors. We believe that such heuristics would allow both GCPG and HPG to achieve good results in both the four rooms environment and Ms. Pac-man. Furthermore, whereas hindsight experience replay is directly applicable to state-of-the-art techniques, our work can probably benefit from being extended to state-of-the-art policy gradient approaches, which we intend to explore in future work. Similarly, we believe that additional heuristics and careful hyperparameter settings would allow DQN+HER to achieve good results in the FetchPush environment. This is evidenced by the fact that Andrychowicz et al. (2017) achieve good results using the deep deterministic policy gradient (Lillicrap et al., 2016, DDPG) in a similar environment (with a continuous action space and a different reward function). The empirical comparisons between either GCPG and HPG or DQN and DQN+HER are comparatively more conclusive, since the similarities between the methods minimize confounding factors.

Regardless of these empirical results, policy gradient approaches constitute one of the most important classes of model-free reinforcement learning methods, which by itself warrants studying how they can benefit from hindsight. Our approach is also complementary to previous work, since it is entirely possible to combine a critic trained by hindsight experience replay with an actor that employs hindsight policy gradients. Although hindsight experience replay does not require a correction analogous to importance sampling, indiscriminately adding hindsight transitions to the replay buffer is problematic, which has mostly been tackled by heuristics (Andrychowicz et al., 2017, Sec. 4.5). In contrast, our approach seems to benefit from incorporating all available information about goals at every update, which also avoids the need for a replay buffer.

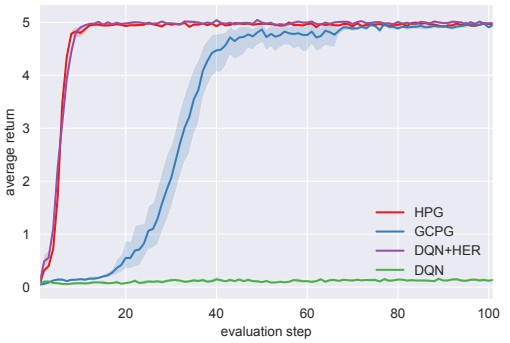

Figure 40: Bit flipping ($k = 8$).

Figure 41: Bit flipping ($k = 16$).

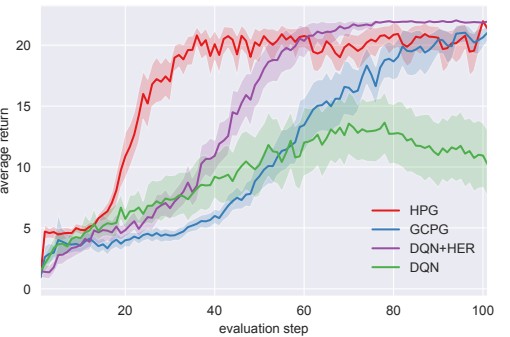

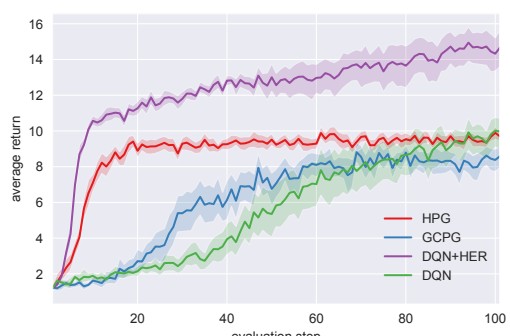

Figure 42: Empty room.

Figure 43: Four rooms.

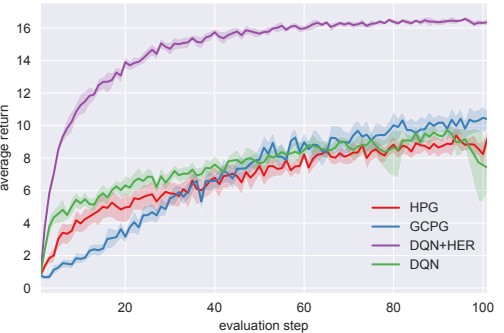

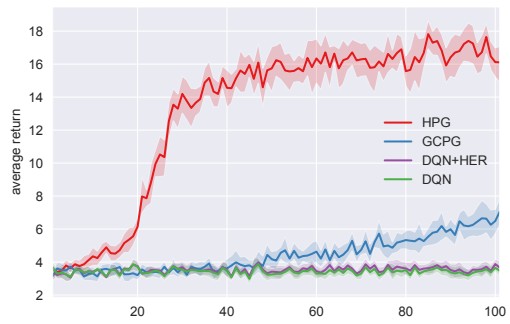

Figure 44: Ms. Pac-man.

Figure 45: FetchPush.

