# OpenReview forum: "Hindsight policy gradients"
_ICLR.cc/2019/Conference_

### Official Review · AnonReviewer2 · 2018-11-02
**A comprehensive consideration of hindsight for policy gradients**

**Rating:** 7
**Confidence:** 4

**Review:**

Following recent work on Hindsight Experience Replay (Andrychowicz et al. 2017), the authors extend the idea to policy gradient methods. They formally describe the goal-conditioned policy gradient setup and derive the extensions of the classical policy gradient estimators. Their key insight to deriving a computationally efficient estimator is that for many situations, only a small number of goals will be "active" in a single trajectory. Then, they conduct extensive experiments on a range of problems and show that their approach leads to improvements in sample efficiency for goal-conditioned tasks.

Although the technical novelty of the paper is not high (many of the estimators follow straightforwardly from previous results, however, the goal subsampling idea is a nice contribution), the paper is well written, the topic is of great interest, and the experiments are extensive and insightful. I expect that this will serve as a nice reference paper in the future, and launching point for future work.

The only major issue I have is that there is no comparison to HER. I think it would greatly strengthen the paper to have a comparison with HER. I don't think it diminishes their contributions if HER outperforms HPG, so I hope the authors can add that.

Comments:

In Sec 6.1, it seems surprising that GCPG+B underperforms GCPG. I understand that HPG+B may underperform HPG, but usually for PG methods a baseline helps. Do you understand what's going on here?

In Sec 6.2, it would be helpful to plot the average return of the optimal policy for comparison (otherwise, it's hard to know if the performance is good or bad). Also, do you have any explanations for why HPG does poorly on the four rooms?

====

Raising my score after the authors responded to my questions and added the HER results.

---

> ### Author Response · Authors · 2018-11-18
> **Re: A comprehensive consideration of hindsight for policy gradients**
>
> Thank you very much for the time that you have dedicated to evaluate our work. We are glad that you believe that our paper is well written and of great interest, that our experiments are extensive and insightful, and that our contribution has the potential to become a reference and starting point for future work.
>
> You are absolutely correct in noting that the fact that only active goals need to be considered is crucial to the feasibility of the proposed estimators. This result is very specific to this application of importance sampling, which also leads to other remarkable properties (as discussed in Section 5). However, we disagree with the claim that the technical novelty of our paper is not high. Firstly, our technical approach to hindsight is radically different from previous work. Secondly, the exact formulation of the hindsight policy gradient, its relationships with value functions, and the feasibility of the corresponding estimators are only clear in hindsight. Finally, although apparently simple by analogy, several results require proofs that are elementary but involved (for an example, see Theorem 4.2).
>
> It is indeed very interesting that including a value function baseline seems more harmful than helpful according to our experiments. After careful investigation, we have concluded that the value function baseline is often poorly fit by the time that the policy exhibits desirable behavior, which is probably due to the fact that the value function baseline is not trained using hindsight. This is particularly evident in the bit flipping environments, the most extreme examples of sparse-reward environments that we consider, where both HPG+B and GCPG+B exhibit unstable behavior (although GCPG+B only ever reaches a good performance for k=8 and a batch of size 16). Although our preliminary experiments in using hindsight to fit a value function baseline have been successful, this may be accomplished in several ways, and requires a careful study of its own.
>
> We believe that the poor performance of every technique in the four rooms environment could be addressed by well-known policy gradient tricks (e.g., entropy bonuses, reward scaling, learning rate annealing, simple statistical baselines), which we have avoided in order to reduce confounding factors in our experiments. The stark state information and the layout that offers a single door between adjacent rooms make this environment surprisingly difficult, but it is probably within reach of agents trained with either HPG or GCPG. Indeed, plotting the average return of the optimal policy would be helpful for inspecting results. We can easily include that in the final version of the paper.
>
> We completely understand your interest in a direct comparison with hindsight experience replay, although we are glad that you agree that our contribution would not be diminished if hindsight experience replay were more sample efficient at this stage. Because this comparison was a common request among reviewers, we are currently working on it. We will provide an updated version of the paper including the corresponding results before the end of the rebuttal period (ideally by 21/11).
>
> Nonetheless, we would like to briefly explain why we did not include such a comparison in the current version of the paper. Firstly, hindsight experience replay is an approach that can be applied to any reinforcement learning technique that relies on experience replay. Besides the choices required to implement hindsight experience replay itself (such as the goal sampling strategy and number of hindsight transitions per observed transition), each of these techniques potentially has several important hyperparameters. Instead of comparing HPG to one of these techniques, we preferred to focus on a rigorous comparison with GCPG, its most natural counterpart. The similarities between both methods allow for a highly systematic comparison that minimizes confounding factors. Secondly, note that we have not used tricks that are known to increase the performance of policy gradient methods, once again in order to avoid introducing confounding factors. Because hindsight experience replay is directly applicable to state-of-the-art techniques, this would lead to an unbalanced comparison. Finally, it should be clear that our work can probably benefit from being extended to state-of-the-art policy gradient approaches. However, once again, such extensions are likely to introduce confounding factors that we would prefer to avoid in our fundamental work.
>
> We hope that these clarifications and the additional experimental content to be released before the rebuttal deadline will allow you to reconsider the rating given to our submission.

---

> > ### Comment · AnonReviewer2 · 2018-11-19
> > **Re: A comprehensive consideration of hindsight for policy gradients**
> >
> > Thank you for your clarifications.
> >
> > That the baseline is so poor is still surprising. A simple running average baseline (that takes into account the number of remaining steps) should do no worse than the original estimator.
> >
> > Look forward to the addition of the HER results. I will raise my score with the updated version of the paper.

---

### Official Review · AnonReviewer3 · 2018-11-02
**AR3: Hindsight policy gradients**

**Rating:** 7
**Confidence:** 4

**Review:**

The authors present HPG, which applies the hindsight formulation already applied to off-policy RL algorithms (hindsight experience replay, HER, Andrychowicz et al., 2017) to policy gradients.
Because the idea is not new, and formulating HPG from PG is so straightforward (simply tie the dynamical model over goals), the work seems incremental. Also, going off policy in PG is known to be quite unstable, and so I'm not sure that simply using the well known approach of normalized importance weights is in practice enough to make this a widely useful algorithm for hindsight RL.


Evaluation      3/5 How does HPG compare to HER? The only common experiment appears to be bit-flipping, which it appears (looking back at the HER paper, no reference to HER performance in this paper) to signifcantly underperform HER. In general I think that the justification for proposing HPG and possible advantages over HER need to be discussed: why should we generalize what is considered an on-policy algorithm like PG to handle hindsight, when HER seems ideally suited for such scenarios? Why not design an experiment that showcases the advantages of HPG over HER?
Clarity              4/5 Generally well explained.
Significance    3/5 The importance of HPG relative to off-policy variants of hindsight is not clear. Are normalized importance weights, a well established variance reduction technique, enough to make HPG highly effective? Do we really want to be running separate policies for all goals? With the practical need to do goal sub-sampling, is HPG really a strong algorithm (e.g. compared to HER)? Why does HPG degrade later in training sometimes when a baseline is added? This is strange, and warrants further investigation.
Originality     2/5 More straightforward extension of previous work based on current presentation.

Overall I feel that HPG is a more straightforward extention of previous work, and is not (yet at least) adequately justified in the paper (i.e. over HER). Furthermore, the experiments seem very preliminary, and the paper needs further maturation (i.e. more discussion about and experimental comparision with previous work, stronger experiments and justification).
Rating          5/10 Weak Reject
Confidence      4/5

Updated Review:

The authors have updated the appendix with new results, comparing against HER, and provided detailed responses to all of my concerns: thank you authors.

While not all of my concerns have been addressed (see below), the new results and discussion that have been added to the paper make me much more comfortable with recommending acceptance. The formuation, while straightforward and not without limitations, has been shown in preliminary experiments to be effective. While many important details (e.g. robust baselines and ultimate performance) still need to be worked out, HPG is almost certainly going to end up being a widely used addition to the RL toolbox. Good paper, recommend acceptance.

Evaluation/Clarity/Originality/Significance: 3.5/4/3/4

Remaining concerns:
- The poor performance of the baselines may indeed be due to lack of hindsight, but this should really be debugged and addressed by the final version of the paper.
- Results throughout the paper are shown for only the first 100 evaluation steps. In many of the figures the baselines are still improving and are highly competitive... some extended results should be included in the final version of the paper (at least in the appendix).
- As pointed out, it is difficult to compare the HER results directly, and it is fair to initially avoid confounding factors, but Polyak-averaging and temporal difference target clipping are important optimization tricks. I think it would strengthen the paper to optimize both the PG and DQN based methods and provide additional results to get a better idea of where things stand on these and/or possibly a more complicated set of tasks.

---

> ### Author Response · Authors · 2018-11-18
> **Re: AR3: Hindsight policy gradients (Part 1/2)**
>
> Thank you very much for the time that you have dedicated to evaluate our work. We are glad that you found our ideas generally well explained.
>
> Regarding your summary of our contribution, although we agree that hindsight is not an original idea in reinforcement learning, it was introduced only recently and has attracted significant interest, as evidenced by the fact that the work of Andrychowicz et al. (2017) has received more than one hundred citations in less than two years, when it first appeared as a technical report.
>
> While we agree that tying the dynamics model over goals is straightforward, that is just one of many steps required to derive our approach, which has importance sampling at its core. Importance sampling is indeed the natural choice to enable using off-policy data in policy gradients. Nonetheless, the exact formulation of the hindsight policy gradient, its relationships with value functions, and the feasibility of the corresponding estimators are only clear in hindsight. For instance, note that we are able to derive an estimator that can be effectively computed for environments of interest even though it seems to require an expectation over all possible goals. Although apparently simple by analogy, several results require proofs that are elementary but involved (for an example, see Theorem 4.2). Our technical approach to hindsight is radically different from previous work, which is why we strongly disagree with the claim that our work is incremental.
>
> The reviewer is correct in noting that employing importance sampling to compute gradients can in general be unstable, which motivates the empirical study presented in Section 6 and the supplementary empirical study of likelihood ratios presented in Appendix E.3.6. We believe that our experiments on a diverse selection of sparse-reward environments conclusively answer the question of whether weighted importance sampling is effective. In addition to such substantial empirical evidence, it is crucial to note that we apply importance sampling in a very specific setting, leading to estimators that have remarkable properties that differentiate them from previous estimators for off-policy learning. We mention several of these properties in Section 5, in the paragraph before the last.
>
> On a related subject, we vehemently disagree with the claim that our experiments are preliminary. Note that Reviewer #2 refers to our experiments as extensive and Reviewer #4 believes that our experiments are well designed and that our analysis is thorough and rigorous.
>
> We completely understand your interest in a direct comparison with hindsight experience replay. Because this comparison was a common request among reviewers, we are currently working on it. We will provide an updated version of the paper including the corresponding results before the end of the rebuttal period (ideally by 21/11).
>
> Nonetheless, we would like to briefly explain why we did not include such a comparison in the current version of the paper. Firstly, hindsight experience replay is an approach that can be applied to any reinforcement learning technique that relies on experience replay. Besides the choices required to implement hindsight experience replay itself (such as the goal sampling strategy and number of hindsight transitions per observed transition), each of these techniques potentially has several important hyperparameters. Instead of comparing HPG to one of these techniques, we preferred to focus on a rigorous comparison with GCPG, its most natural counterpart. The similarities between both methods allow for a highly systematic comparison that minimizes confounding factors. Secondly, note that we have not used tricks that are known to increase the performance of policy gradient methods (e.g., entropy bonuses, reward scaling, learning rate annealing, simple statistical baselines), once again in order to avoid introducing confounding factors. Because hindsight experience replay is directly applicable to state-of-the-art techniques, this would lead to an unbalanced comparison. Finally, it should be clear that our work can probably benefit from being extended to state-of-the-art policy gradient approaches. However, once again, such extensions are likely to introduce confounding factors that we would prefer to avoid in our fundamental work.
>
> (Part 1/2)

---

> > ### Author Response · Authors · 2018-11-19
> > **Re: AR3: Hindsight policy gradients (Part 2/2)**
> >
> > Regardless of the results of the direct empirical comparison with hindsight experience replay that we will provide, several facts justify our work. Firstly, policy gradient approaches constitute one of the most important classes of model-free reinforcement learning methods, which by itself warrants studying how they can benefit from hindsight. Our empirical results show that our approach is more than just theoretically interesting. Because such approach is complementary to previous work, note that it is entirely possible to train a critic by hindsight experience replay while training an actor that employs hindsight policy gradients. Secondly, although hindsight experience replay does not require a correction analogous to importance sampling, indiscriminately adding hindsight transitions to the replay buffer is problematic, which has mostly been tackled by heuristics (see Andrychowitz et al. (2017), Sec. 4.5). In contrast, our approach seems to benefit from incorporating all available information about goals at every update, which also avoids the need for a memory-costly replay buffer. The practical need for active goal subsampling in longer episodes seems to lead to a natural trade-off between computational efficiency and sample efficiency. Note that many successful model-free reinforcement learning algorithms rely on an approximation to a principled formulation, and our experiments suggest that subsampling active goals is such an example.
> >
> > Although the bit flipping and FetchPush environments used in our evaluation are similar to environments found in the work of Andrychowicz et al. (2017), our results are not directly comparable to theirs, mainly due to differences in the evaluation protocol. The most significant of such differences is that Andrychowicz et al. (2017) are only concerned with whether a goal was achieved during an episode, whereas we are also concerned with whether a goal was achieved quickly.
> >
> > Regarding the fact that policies sometimes degrade during training with HPG+B, first note that this phenomenon is only observed in bit flipping environments. Also note that GCPG+B presents instability in the only bit flipping environment where it does not perform poorly (batch size 16, k = 8). After careful investigation, we are convinced that the main cause of this issue is the fact that the value function baseline is still very poorly fit by the time that the policy exhibits desirable behavior. In all likelihood, this is due to the fact that the value function baseline is not trained using hindsight, which is also consistent with the fact that the instability is observed precisely in the the most extreme examples of sparse-reward environments. Although our preliminary experiments in using hindsight to fit a value function baseline have been successful, this may be accomplished in several ways, and requires a careful study of its own.
> >
> > Besides the additional experimental content to be released before the rebuttal deadline, we can easily improve the justification for our approach by including the arguments presented above in the final version of the paper. We hope that these changes will allow you to reconsider the rating given to our submission.
> >
> > (Part 2/2)

---

### Official Review · AnonReviewer4 · 2018-11-10
**Formalizes hindsight experience replay as importance sampling in policy gradients. Good paper with clear contribution despite low novelty.**

**Rating:** 7
**Confidence:** 4

**Review:**

This paper extends the work of Hindsight Experience Replay to (goal-conditioned) policy gradient methods. Hindsight, which allows one to learn policies conditioned on some goal g, from off-policy experience generated by following goal g’, is cast in the framework of importance sampling. The authors show how one can simply rewrite the goal-conditioned policy gradient by first sampling a trajectory, conditioned on some goal $g’$ and then computing the closed form gradient in expectation over all goals. This gradient is unbiased if the rewards are off-policy corrected along the generated trajectories. While this naive formulation is found to be unstable , the authors propose a simple normalized importance sampling formulation which appears to work well in practice. To further reduce variance and computational costs, the authors also propose goal subsampling mechanisms, which sample goals which are likely along the generated trajectories. The method is evaluated on the same bit-flipping environment as [1], and a variety of discrete environments (grid worlds, Ms. Pac-Man, simulated robot arm) where the method appears highly effective. Unfortunately for reasons which remain unclear, hindsight policy gradients with value baselines appear unstable.

Quality:
This paper scores high wrt. quality. The theoretical contributions of the method are solid, the experiments are well designed and highlight the efficacy of the method, as well as areas for improvement. In particular, I commend the authors for the rigorous analysis (bootstrapped error estimates, separate seeds for hyper-parameters and reporting test error, etc.), including the additional results found in the appendix (sensitivity and ablative analyses). That being said, the paper could benefit from experiments in the continuous control domain and a direct head-to-head comparison with HER. While I do not anticipate the proposed method to outperform HER in terms of data-efficiency (due to the use of replay) the comparison would still be informative to the reader.

Clarity:
The paper is well written and easy to follow. If anything, the authors could have abridged sections 2 and 3 in favor of other material found in the Appendix, as goal-conditioned policy gradients (and variants) are straightforward generalizations of standard policy gradient methods.

Originality:
Novelty is somewhat low for the paper as Hindsight Experience Replay already presented a very similar off-goal-correction mechanism for actor-critic methods (DDPG). The method is also very similar to [2], the connection to which should also be discussed.

Significance.
Despite the low novelty, I do believe there is value in framing “hindsight” as importance sampling in goal-conditioned policy gradients. This combined with the clear presentation and thorough analysis in my opinion warrants publication and will certainly prove useful to the community. Significance could be improved further should the paper feature a more prominent discussion / comparison to HER, along with a fix for the instabilities which occur when using their method in conjunction with a value baseline.

[1] Hindsight Experience Replay. Marcin Andrychowicz et al.
[2] Data-Efficient Hierarchical Reinforcement Learning. Ofir Nachum, Shixiang Gu, Honglak Lee, Sergey Levine.

Detailed Comments:
* Section 2: “this formulation allows the probability of a state transition given an action to change across time-steps within an episode”. I do not understand this statement, as $p(s_{t+1} \mid s_t, a_t)$ is the same transition distribution found in standard MDPs, and appears stationary wrt. time.
* Theorems 3.1 - 3.1 (and equations). A bit lengthy and superfluous. Consider condensing the material.
* Section 5: I found the change in notation (from lower to upper-case) somewhat jarring. Also, the notation used for empirical samples from the mini-batch is confusing. If $A^{(i)}_t}$ is meant to be the action at time-step $t$ for the $i$-th trajectory in the minibatch, then what does $G^{(i)} = g$ mean ? I realize this means evaluating the probability by setting the goal state to $g$, but this is confusing especially when other probabilities are evaluated conditioned on $G^{(i)}$ directly.
* Section 6. “Which would often require the agent to act after the end of an episode”. Do you mean that most episodes have length T’ < T, and as such we would “waste time” generating longer trajectories ?
* RE: Baseline instabilities. Plotting the loss function for the value function could shed light on the instability.

---

> ### Author Response · Authors · 2018-11-18
> **Re: Formalizes hindsight experience replay as importance sampling in policy gradients (...) (Part 1/2)**
>
> Thank you very much for the time that you have dedicated to evaluate our work. We are glad that you believe that our work is high quality, that it will be useful to the community, that our paper is well written, that our theoretical contributions are solid, that our experiments are well designed, that our experimental analysis is rigorous, that our hyperparameter sensitivity and ablation analyses are valuable, and that our method appears highly effective.
>
> Regarding your brief summary of our work, although this is probably clear, we would like to emphasize that restricting attention to active goals does not affect the HPG estimator (a remarkable property), while subsampling active goals increases computational efficiency likely at a cost in sample efficiency. Therefore, both strategies are not intended to reduce variance.
>
> We completely understand your interest in a direct comparison with hindsight experience replay, although we are glad that you agree that our contribution would not be diminished if hindsight experience replay were more sample efficient at this stage. Because this comparison was a common request among reviewers, we are currently working on it. We will provide an updated version of the paper including the corresponding results before the end of the rebuttal period (ideally by 21/11).
>
> Nonetheless, we would like to briefly explain why we did not include such a comparison in the current version of the paper. Firstly, hindsight experience replay is an approach that can be applied to any reinforcement learning technique that relies on experience replay. Besides the choices required to implement hindsight experience replay itself (such as the goal sampling strategy and number of hindsight transitions per observed transition), each of these techniques potentially has several important hyperparameters. Instead of comparing HPG to one of these techniques, we preferred to focus on a rigorous comparison with GCPG, its most natural counterpart. The similarities between both methods allow for a highly systematic comparison that minimizes confounding factors. Secondly, note that we have not used tricks that are known to increase the performance of policy gradient methods (e.g., entropy bonuses, reward scaling, learning rate annealing, simple statistical baselines), once again in order to avoid introducing confounding factors. Because hindsight experience replay is directly applicable to state-of-the-art techniques, this would lead to an unbalanced comparison. Finally, it should be clear that our work can probably benefit from being extended to state-of-the-art policy gradient approaches. However, once again, such extensions are likely to introduce confounding factors that we would prefer to avoid in our fundamental work.
>
> We plan on conducting and presenting experiments on environments with continuous action spaces in future work.
>
> Regarding the fact that policies sometimes degrade during training with HPG+B, first note that this phenomenon is only observed in bit flipping environments. Also note that GCPG+B presents instability in the only bit flipping environment where it does not perform poorly (batch size 16, k = 8). After careful investigation, we are convinced that the main cause of this issue is the fact that the value function baseline is still very poorly fit by the time that the policy exhibits desirable behavior. In all likelihood, this is due to the fact that the value function baseline is not trained using hindsight, which is also consistent with the fact that the instability is observed precisely in the the most extreme examples of sparse-reward environments. Although our preliminary experiments in using hindsight to fit a value function baseline have been successful, this may be accomplished in several ways, and requires a careful study of its own.
>
> Although we agree that Section 2 and Section 3 could be abridged in favour of results presented in the Appendices, we also believe that these two sections make the paper more accessible. For instance, Section 4 certainly benefits from the previous presentation of goal-conditional counterparts of well known results (Section 3) using our notation (Section 2).
>
> We disagree with the claim that our paper is somewhat less novel because hindsight experience replay may be applied to the deep deterministic policy gradient (DDPG). Firstly, the correction mechanism based on importance sampling that we propose is radically different in comparison to the approach based on experience replay. Secondly, and more importantly, the deep deterministic policy gradient is not in the same class of policy gradient algorithms that we consider, which contains important state-of-the-art algorithms. Note that DDPG requires a critic that is differentiable with respect to the choice of action by the actor (for any state). Consequently, the method only applies to environments with continuous action spaces.
>
> (Part 1/2)

---

> > ### Author Response · Authors · 2018-11-19
> > **Re: Formalizes hindsight experience replay as importance sampling in policy gradients (...) (Part 2/2)**
> >
> > We thank the reviewer for suggesting the work of Nachum et al. (2018). Their work seems related to the work of Levy et al. (2017), which applies hindsight experience replay in a hierarchical reinforcement learning approach.
> >
> > We address your detailed comments in order of appearance:
> > * We note that $p(s_{t+1} \mid s_t, a_t)$ may be different from $p(s_{t'+1} \mid s_{t'}, a_{t'})$ even if $s_{t+1} = s_{t'+1}$, $s_{t} = s_{t'}$, and $a_{t} = a_{t'}$, as long as $t \neq t'$. Following the convention briefly introduced in the first paragraph of Section 2, $p(s_{t+1} \mid s_{t}, a_{t})$ refers to the conditional probability of $S_{t+1} = s_{t+1}$, whereas $p(s_{t'+1} \mid s_{t'}, a_{t'})$ refers to the conditional probability of $S_{t'+1} = s_{t'+1}$. Because $S_{t+1}$ and $S_{t'+1}$ are distinct random variables when $t \neq t'$, our task formulation allows the probability of a state transition given an action to change across time steps within an episode. This is reminiscent of the notation employed by Bishop (2013). Our notation would also allow a different policy for each time step, which is why we have noted that the same policy is used to make decisions at every time step. We understand that this may be confusing at first, but we have been very careful in our choice of notation to allow for conciseness and rigour.
> > * As discussed previously, we believe that Section 3 is important to make the paper as a whole more accessible.
> > * Because estimators are random variables, they are indeed written as functions of other random variables (instead of their realizations). In your example, $G^{(i)} = g$ refers to conditioning the random variable $G^{(i)}$ on the realization $g$, whereas a probability conditioned on $G^{(i)}$ will be a function of a random variable (itself a random variable). In any case, the reviewer seems to have interpreted the expression correctly.
> > * That is correct. Note that interacting with the environment after a particular goal has been achieved would provide an HPG agent with additional information about alternative goals.
> > * As discussed previously, we have already investigated the instability of value function baselines in bit flipping environments.
> >
> > We hope that these clarifications and the additional experimental content to be released before the rebuttal deadline will allow you to reconsider the rating given to our submission.
> >
> > (Part 2/2)

---

### Public Comment · (anonymous) · 2018-10-02
**Gradients being zero when rewards are sparse**

If I understand this correctly, if the rewards are sparse i.e if goal reached then reward is 1 else 0, wouldn't be your gradient 0 most of the time in equation 6? If that is the case, what is the need of the importance sampling then?

---

> ### Author Response · Authors · 2018-10-03
> **Re: Gradients being zero when rewards are sparse**
>
> Thank you for your interest. Note that Equation 6 involves rewards computed for every possible goal, not only for an original goal. Therefore, as long as there is an (alternative) goal for which a trajectory obtains a non-zero reward, the corresponding term will probably be non-zero. Section 5 details how the corresponding estimator can be computed efficiently.

---

### Author Response · Authors · 2018-11-21
**Update: comparison between hindsight policy gradients and hindsight experience replay**

We have updated the paper to include an empirical comparison between hindsight policy gradients and hindsight experience replay. This comparison is presented in Appendix E.3.7 (Pgs. 36-38).

---

### Meta-Review · Area_Chair1 · 2018-12-15
**A straightforward but solidly useful contribution.**

**Confidence:** 5
**Recommendation:** Accept (Poster)

**Metareview:**

The paper generalizes the concept of "hindsight", i.e. the recycling of data from trajectories in a goal-based system based on the goal state actually achieved, to policy gradient methods.

This was an interesting paper in that it scored quite highly despite all three reviewers mentioning incrementality or a relative lack of novelty. Although the authors naturally took some exception to this, AC personally believes that properly executed, contributions that seem quite straightforward in hindsight (pun partly intended) can be valuable in moving the field forward: a clean and didactic presentation of theory backed by well-designed and extensive empirical investigation (both of which are adjectives used by reviewers to describe the empirical work in this paper) can be as valuable, or moreso, than a poorly executed but higher-novelty works. To quote AnonReviewer3, "HPG is almost certainly going to end up being a widely used addition to the RL toolbox".

Feedback from reviewers prompted extensive discussion and a direct comparison with Hindsight Experience Replay which reviewers agreed added significant value to the manuscript, earning it a post-rebuttal unanimous rating of 7. It is therefore my pleasure to recommend acceptance.